# Carbon isotopic ratios of modern C3 and C4 vegetation on the Indian Peninsula and changes along the plant–soil–river continuum; implications for vegetation reconstructions

Frédérique M.S.A. Kirkels[1], Hugo J. de Boer[2], Paulina Concha Hernández[1], Chris R.T. Martes[1], Marcel T.J. van der Meer[3], Sayak Basu[4,a], Muhammed O. Usman[5,b], Francien Peterse[1]

[1]Department of Earth Sciences, Faculty of Geosciences, Utrecht University, Princetonlaan 8a, 3584 CB Utrecht, The Netherlands

[2]Department of Environmental Sciences, Copernicus Institute of Sustainable Development, Faculty of Geosciences, Utrecht University, Princetonlaan 8a, 3584 CB Utrecht, The Netherlands

[3]Department of Marine Microbiology and Biogeochemistry, NIOZ Royal Netherlands Institute for Sea Research, PO Box 59, 1790 AB Den Burg, the Netherlands

[4]Department of Earth Sciences, Indian Institute of Science Education and Research Kolkata, Mohanpur 741246, India

[5]Geological Institute, ETH Zürich, Sonneggstrasse 5, 8092 Zürich, Switzerland

[a]Present address: Geological Oceanography Department, National Institute of Oceanography, Dona Paula - 403 004, Goa, India

[b]Present address: Department of Physical & Environmental Sciences, University of Toronto Scarborough, Toronto, Ontario M1C1A4, Canada

*Correspondence to*: Francien Peterse (f.peterse@uu.nl) and Frédérique Kirkels (f.m.s.a.kirkels@uu.nl)

**Abstract**

The large difference in the fractionation of stable carbon isotopes between C3 and C4 plants is widely used in vegetation reconstructions, where the predominance of C3 plants suggests wetter and that of C4 plants drier conditions. The stable carbon isotopic composition of organic carbon (OC) preserved in soils or sediments may be a valuable (paleo-)environmental indicator, based on the assumption that plant-derived material retains the stable carbon isotopic value of its photosynthetic pathway during transfer from plant to sediment. In this study, we investigated the bulk carbon isotopic values of C3 and C4 plants ($\delta^{13}C$) and of organic carbon ($\delta^{13}C_{org}$) in soils, river Suspended Particulate Matter (SPM) and riverbed sediments, to gain insight in the control of precipitation on C3 and C4 plant $\delta^{13}C$ values and to assess changes in $\delta^{13}C_{org}$ values along the plant–soil–river continuum. This information allows us to elucidate the implications of different $\delta^{13}C$ end-members on C3/C4 vegetation reconstructions. Our analysis was performed in the Godavari River basin, located in the Core Monsoon Zone in peninsular India, a region that integrates the hydroclimatic and vegetation changes caused by variation in monsoonal strength. The basin has distinct wet and dry seasons and is characterised by natural gradients in soil type (from clay-rich to sandy), precipitation (~500 to 1500 mm y$^{-1}$) and vegetation type (from mixed C3/C4 to primarily C3) from the upper to the lower basin. The $\delta^{13}C$ values of Godavari C3 plants were strongly controlled by Mean Annual Precipitation (MAP), showing an isotopic enrichment of ~2.2 ‰ from ~1500 to 500 mm y$^{-1}$. Tracing $\delta^{13}C_{org}$ values from plant to soils and rivers revealed that soils and riverbed sediments reflected the transition from mixed C3 and C4 vegetation in the dry upper basin to more C3 vegetation in the humid lower basin. Soil degradation and stabilisation processes and hydrodynamic sorting within the river altered the plant-derived $\delta^{13}C$ signal. Phytoplankton dominated the $\delta^{13}C_{org}$ signal carried by SPM in the dry season and year-round in the upper basin. Application of a linear mixing model showed that the %C4 plants in the different subbasins was ~7–15 % higher using plant end-members based on measurement of the Godavari vegetation and tailored to local moisture availability than using those derived from data compilations of global vegetation. Including a correction for the $^{13}$C-enrichment in Godavari C3 plants due to drought resulted in maximal 6 % lower estimated C4 plant cover. Our results from the Godavari basin underline the importance of making informed choices about the plant $\delta^{13}C$ end-members for vegetation reconstructions, considering characteristics of the regional vegetation and environmental factors such as MAP in monsoonal regions.

## 1. Introduction

Vegetation reconstruction of the coverage of C3 and C4 plants uses the distinct $\delta^{13}C$ values of both vegetation types with the assumption that Organic Carbon (OC) retains the stable carbon isotopic composition during transfer from plant to soils and sediments (e.g., Koch, 1998; Dawson et al., 2002; Wynn and Bird, 2007). The C3 and C4 photosynthetic pathways fractionate carbon isotopes to a different extent; this is reflected in bulk $\delta^{13}C$ values of ~-20 to -37 ‰ in C3 and ~-10 to -16 ‰ in C4 plants (e.g., Bender, 1971; Farquhar et al., 1989; Kohn, 2010). C3 plants fix $CO_2$ using the Calvin–Benson cycle and they are prevalent in relatively cold and humid environments. C4 plants add an initial $CO_2$ fixation step using PEP carboxylase to concentrate $CO_2$ via bundle sheath cells inside the leaf. This additional step in the photosynthesis pathway allows C4 plants to maintain relatively high photosynthesis rates under low stomatal conductance with limited water loss, which enables them to thrive in high temperature and (semi-)arid environments (e.g., Farquhar, 1983; Sage and Monson, 1999; Sage, 2004).

Given the sensitivity of vegetation type to water availability, shifts in the relative contribution of C3 and C4 plants can be used to infer hydroclimatic changes, with a shift to more dominant C4 vegetation in drier periods or areas and more dominant C3 vegetation in wetter conditions (Koch, 1998; Sage, 2004; Still et al., 2003 and references therein). The bulk carbon isotopic composition of organic matter ($\delta^{13}C_{org}$) preserved in soils or in river-dominated sediments in the marine realm is often used as proxy for vegetation reconstructions as it is considered to represent an integrated signal of the vegetation (e.g., Galy et al., 2007; Sarangi et al., 2021). For vegetation reconstructions, different techniques can be employed, including bulk or compound-specific isotope analyses. Although less source specific, bulk isotope analyses provide a low-cost, high throughput approach that can be applied at high resolution and/or large geographic areas, also in (sub-)tropical regions where carbon and vegetation-specific compound concentrations are generally low and may undergo compound-specific degradation patterns and/or differential settling into sediments (e.g., Hou et al., 2020; Li et al., 2020). Subsequently, isotope mixing models provide a means to infer the distribution of C3 and C4 plants and changes therein at spatial or temporal scales, which is particularly relevant in context of changing climatic conditions affecting C3/C4 vegetation patterns. For instance, gradual aridification in a basin can result in drought-stressed C3 plants, as well as increased abundance of aridity-adapted C4 plants. Both changes result in an increase in

bulk $\delta^{13}C_{org}$ values. A shift to more dominant C4 vegetation is important to identify as a shift to a dry ecosystem indicates a reduced resilience to changes in moisture availability (e.g., Cui et al., 2017; Ghosh et al., 2017).

The monsoon-influenced Indian subcontinent is particularly sensitive to changes in hydroclimate on both short (seasonal) and long (orbital) timescales (Turner and Annamalai, 2012; Sinha et al., 2011, 2015; Banerji et al., 2020; Dutt et al., 2021), resulting in changes in the C3 and C4 vegetation distributions over the Neogene and Quaternary period (e.g., Agrawal et al., 2012; Ghosh et al., 2017; Basu et al., 2018, 2019a; Roy et al., 2020). For example, changes in bulk $\delta^{13}C_{org}$ and leaf wax-specific $\delta^{13}C$ values captured the late Miocene (7.4–7.2 Ma)

expansion of C4 plants recorded in Himalayan-derived Indus fan sediments (Feakins et al., 2020) as well as in Gangetic plain alluvial sediments and paleosols (Ghosh et al., 2017; Roy et al., 2020). C4 plants spread southward over the Indian peninsula during the mid-Pliocene to mid-Pleistocene (3.5–1.5 Ma) linked to reduced rainfall (Dunlea et al., 2020), whereas an increase in monsoon strength over the last deglaciation led to a shift to a more C3-dominated ecosystem (e.g. Galy et al., 2008a; Contreras-Rosales et al., 2014). Finally, an increase in

C4 plants across India from the mid- to late Holocene, establishing the modern-day vegetation, was linked to aridification (Ponton et al., 2012; Contreras-Rosales et al., 2014; Sarkar et al., 2015; Usman et al., 2018; Basu et al., 2019a). At seasonal to decadal scale, changes in monsoon intensity and distribution can affect plant $\delta^{13}C$ as well as preservation and provenance of the bulk $\delta^{13}C_{org}$ signal in soils and sediments (e.g. Ittekkot et al., 1985; Galy et al., 2008b). However, the effect of monsoon variation on the bulk $\delta^{13}C_{(org)}$ signal along the plant–soil–

river continuum has not been thoroughly tested for peninsular India.

     One approach to reconstruct vegetation is the application of a straightforward, linear isotope mixing model, which requires an informed choice about the C3 and C4 plant $\delta^{13}C$ end-members and their variability that are used as input. These end-members can be based on averages determined in compilations of vegetation occurring

around the globe or be based on (modern) vegetation samples in a region, but also their date of sampling with respect to OC turnover rates in soils or sediments as well as controls by environmental factors need to be considered. Meta-analyses of global C3 vegetation revealed that hydroclimatic conditions such as rainfall amount and seasonality affect the plant $\delta^{13}C$ (Diefendorf et al., 2010; Kohn, 2010; Basu et al., 2019b, 2021). In particular drought stress results in a less negative $\delta^{13}C$ in C3 plants, where mean annual precipitation (MAP) has

a much stronger control than other environmental factors such as temperature or altitude (Stewart et al., 1995; Diefendorf et al., 2010). Observed changes in C3 plant $\delta^{13}C$ due to drought stress are the result of changes in the

ratio of leaf interior to atmospheric $CO_2$ concentrations, which can be the result of changes in stomatal conductance, photosynthetic capacity and photosynthetic rate, or a combination thereof (Farquhar et al., 1989; Diefendorf et al., 2010; Liu et al., 2013). For C4 vegetation, carbon isotope fractionation is generally unaffected by drought stress, although a few field and experimental studies have shown that C4 plant $\delta^{13}C$ becomes more negative under water-limiting conditions as a result of a less efficient $CO_2$ concentrating mechanism referred to as 'bundle sheath leakiness' (e.g., Buchmann et al., 1996; Yoneyama et al., 2010; Basu et al., 2015; Ellsworth and Cousins, 2016). Plants using an alternative photosynthetic pathway i.e., crassulacean acid metabolism (CAM) as adaption to aridity photosynthesise during the day and respire at night (i.e., temporal $CO_2$ concentrating mechanism), where moisture availability determines the expression of the C3 or C4 fixation pattern (Sankhla et al., 1975). Under water-stressed conditions CAM plants have stable carbon isotopic values similar to C4 plants, but they are relatively rare in India (Sankhla et al., 1975; Ziegler et al., 1981) and are, therefore, not further considered here. Regardless, drought stress has the largest impact on C3 plants and is recognised to cause high intraspecies variability in water-limited ecosystems (<1000 mm y$^{-1}$; e.g. Ma et al., 2012; Liu et al., 2013, 2014; Luo et al., 2021). Hence, the existing plant community in a region and the impact of water availability on those plant species may vary locally and result in a C3 plant $\delta^{13}C$ value that differs depending on the regional conditions (Liu et al., 2014; Basu et al., 2019b).

Although it is possible to determine region-specific plant $\delta^{13}C$ end-members that are representative of the regional conditions, including vegetation species, structure, density, agriculture/land-use and important environmental controls such as MAP, this approach requires detailed knowledge of the $\delta^{13}C$ values of the regional C3 and C4 vegetation as well as of rainfall distributions. Problematically, such detailed information is often unavailable and yet to be established for Indian plants in the Core Monsoon Zone (CMZ). Furthermore, correction of the plant end-members for drought conditions that (seasonally) prevail in peninsular India requires details on regional rainfall distributions and depends on the $\delta^{13}C$ value that derived from measurement of regional vegetation or estimated based on data compilations of global vegetation. Recently, a study of $\delta^{13}C$ values in region-specific vegetation along a precipitation gradient on the Gangetic plain prompted a recalculation of the abundance C3 and C4 plants in sedimentary deposits accounting for drought-stress induced $^{13}C$-enrichment in C3 plants (Basu et al., 2015, 2019b). They showed that earlier investigations likely underestimated the abundance of C4 plants (~20 %). Recalculation using an end-member and mixing model approach revealed that C4 plants existed in this region at an earlier date than anticipated, changing the timing of

(Miocene) C4 grassland expansion on the Gangetic plain to ~17 Ma (Basu et al., 2015, 2019b). This shift highlights the effect of plant $\delta^{13}$C end-member values on vegetation reconstructions.

Next to precipitation controls on vegetation $\delta^{13}$C, the initial plant $\delta^{13}$C signal may be altered during transit from plant to the sedimentary archive, depending on physical and biogeochemical processes that determine the stability i.e., protection against degradation and transport efficiency of this plant-derived OC (e.g., Battin et al., 2009; Ward et al., 2017). Soil and sedimentary deposits integrate a temporal signal, depending on OC turnover rates which are estimated to range from ~10 years in tropical forest soils to ~25 – 40 years in savanna soils

(Martin et al., 1990; Bird et al., 1996). Comparison of the older $\delta^{13}$C value of soils and sediment with that of the modern vegetation, requires consideration of the Suess effect that describes the rapid decline over the last few decades in the $\delta^{13}$C value of atmospheric $CO_2$ as a result of fossil fuel burning, causing a change in the OC stable carbon isotopic composition of vegetation over time. Furthermore, it is well-established that soil degradation processes lead to a $^{13}$C-enrichment in the remaining soil OC, which is usually estimated to be ~1– 3

160    ‰ but can be as high as 6 ‰ in tropical and semi-arid regions (e.g., Krull et al., 2005). Possible factors that contribute to this $^{13}$C-enrichment are preferred uptake and degradation to $CO_2$ of $^{13}$C-depleted OC by microbes, incorporation of $^{13}$C-enriched microbial and fungal biomass in the soil and/or preferential adsorption of $^{13}$C by fine mineral particles (Krull et al., 2005; Wynn, 2007; Wynn and Bird, 2007). Soil OC thus comprises a mixture of plant-derived, fungal and bacterial biomass and microbially processed carbon. The different compounds (e.g.,

lipids, proteins, carbohydrates, etc.) differ in their degradability, but may also be associated with mineral surfaces, which protects them from degradation. Compound-specific degradation rates or preservation in soils via microbial processing or association with mineral particles may influence the reconstructed C3/C4 vegetation balance depending on the targeted compound, as shown for vegetation and soils in the Gangetic plain (Sarangi et al., 2021; Roy and Sanyal, 2022). This complex interplay between different inputs and microbial processing

may challenge the use of stable carbon isotope ratios for vegetation reconstructions.

Furthermore, the marked hydrological changes in Indian monsoonal rivers can change the source and thereby the $\delta^{13}$C value of the OC that it contains at a seasonal scale, from mainly soil-derived OC in the wet season to aquatic produced OC in the dry season, or change its provenance by sourcing from particular parts of the basin

with a different vegetation cover in response to the rainfall distribution (Gupta et al., 1997; Balakrishna and Probst, 2005; Aucour et al., 2006; Galy et al., 2008b, 2011; Kirkels et al., 2020a; Menges et al., 2020). For

example, Galy et al. (2008b) showed that $\delta^{13}C_{org}$ values of suspended particulate matter (SPM) in the Ganges-Brahmaputra River reflected dominant C3 input in the Himalayan tributaries, but after in-river degradation, this signal was replaced by C4 inputs in the Gangetic plain. Finally, hydrodynamic sorting within the river may result in depth-specific OC distributions and thereby influence the $\delta^{13}C$ signal that is transported downriver (Galy et al., 2008b; Bouchez et al., 2014; Feng et al., 2016; Repasch et al., 2022). Hence, interpretation of $\delta^{13}C$-based vegetation reconstructions needs to consider potential alterations during transit from plant source to sedimentary deposits.

In this study, we examine the $\delta^{13}C$ values of C3 and C4 vegetation in the modern-day Godavari River basin, the largest monsoonal river of peninsular India (Fig. 1a), to examine links between monsoon-driven hydroclimate and plant $\delta^{13}C$ values. In addition, we analyse the bulk $\delta^{13}C_{org}$ values in soils, river SPM and riverbed sediments collected in a wet and dry season to explore the evolution of the initial plant-derived $\delta^{13}C$ signal along the plant–soil–river continuum. Finally, we use our insights in the modern system to assess the influence of drought stress and the use of region-specific plant $\delta^{13}C$ end-members on C3/C4 vegetation mixing model estimates and thus the uncertainty of $\delta^{13}C$-based vegetation reconstructions.

## 2. Materials and methods

### 2.1. Regional Setting

The Godavari is the largest peninsular river of India (catchment area: $3.1*10^5$ km$^2$, length: 1465 km) with an annual discharge of 110 km$^3$ and sediment load of 170 Mt of which ~2.8 Mt of OC (Biksham and Subramanian, 1988a,b; Gupta et al., 1997). The Godavari River starts in the Western Ghats mountains and flows across peninsular India before emptying in the Bay of Bengal, and is situated in the Core Monsoon Zone (Ponton et al., 2012; Sarkar et al., 2015; Giosan et al., 2017) (Fig. 1a,b), which dictates the seasonality of the Godavari River, with 75–85% of the annual rainfall and 98 % of the sediment transport in the monsoon/wet season between June and September (Biksham and Subramanian, 1988a,b).

The basin is characterized by several natural gradients, where the upper basin developed on Deccan flood basalts which weathered into clay-rich soils, while the lower basin formed on felsic rock formations with sandy to loamy textured soils (Giosan et al., 2017). Petrogenic OC is absent in the upper basin (Reddy et al., 2021) and

very sporadic (i.e., coal deposits) in the lower basin (Usman et al., 2018). In addition, the precipitation gradient ranges from ~430 mm y$^{-1}$ in the interior upper basin that is in the rain shadow of the Western Ghats mountain range, to ~2300 mm y$^{-1}$ in near the Bay of Bengal coast (Fig. 2). The natural vegetation reflects this gradient and

varies from (C4) grasses, dry deciduous forests and thorny shrublands in the upper basin to moist and evergreen deciduous forests with mostly C3 flora in the lower basin (Olson et al., 2001; Asouti and Fuller, 2008; Fig. 1b). Agriculture covers ~60 % of the basin with dominant C4 crops (sorghum, millet, maize, sugar cane) in the upper basin and rice fields (C3) in the lower basin (CWC, 2014; Pradhan et al., 2014). Hence, the upper and lower Godavari basin are distinctly different in terms of C3/C4 vegetation distributions, moisture conditions and

bedrock geology.

The Godavari basin is divided in 5 subbasins: the Upper (~37 % of the total basin area), Middle (6 %) and Lower (2 %) Godavari cover the main stem river, and are joined by the North (35 %; Wainganga, Penganga, Wardha and Pranhita rivers) and East Tributaries (20 %; Indravati and Sabari rivers) (Babar and Kaplay, 2018)

(Fig. 1b, S1). Abundant dams in the upper basin limit the river flow, while a large dam with reservoir lake at Rajahmundry controls the flow into the tidally influenced delta (Pradhan et al., 2014).

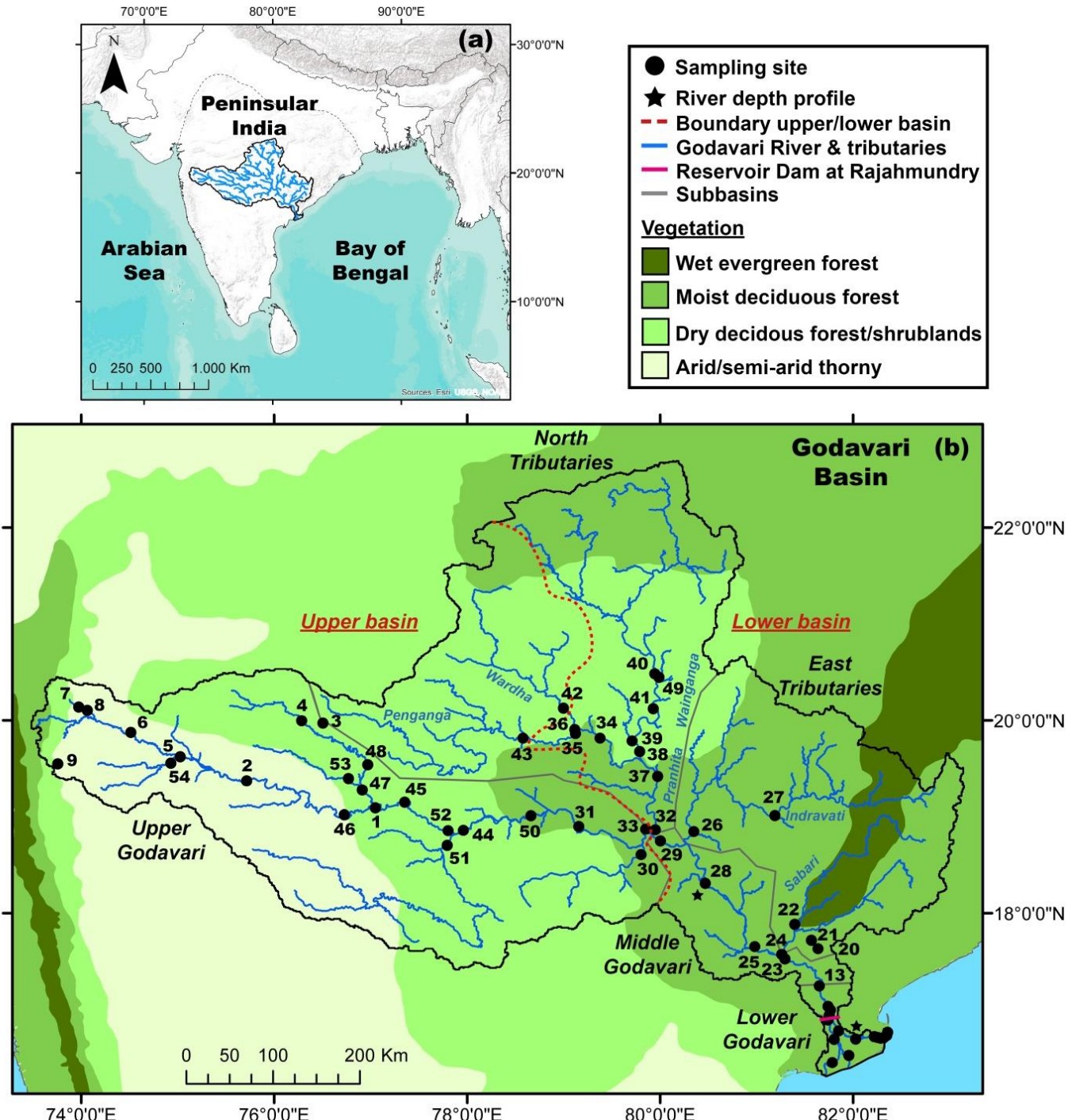

**Fig. 1: (a)** Location of the Godavari River basin in peninsular India. **(b)** Godavari River basin and sampling sites, with the major vegetation zones (Olson et al., 2001; Asouti and Fuller, 2008). Names of the subbasins (grey) and major rivers (blue) are indicated, a zoom for the Godavari delta is available in Fig. S1.

### 2.2. Sample collection

Samples of above-ground plant material were collected in February/March 2015 (dry season) across the Godavari basin, selecting the 3–5 most dominant species at each site and spanning the full range of plant lifeforms (i.e., trees, shrubs, herbaceous plants and grasses). For shrubs and trees the leaves were collected and for herbs and grasses the leaves and stems were combined. Given that the deciduous trees and shrubs shed their leaves annually in the dry season (Kushwaha and Singh, 2005; Elliott et al., 2006), leaves were considered the main contributor to soil OC rather than woody biomass. Depending on plant size, each sample consisted of approximately 10 to 50 grams of leaves or aboveground plant parts of 3–5 individuals of the same species or multiple 'sun' and 'shade' leaves of the same individual. Dominant agricultural crops (e.g. sorghum, maize, millet, sugarcane) were also sampled. After collection, plant samples were air-dried and subsequently frozen upon arrival in the laboratory. In total, 77 samples of C3 plants and 16 samples of C4 plants were prepared for analysis.

Topsoils (1–10 cm) were collected during the same campaign (n=47), after removal of the litter layer and by combining 3–5 spatial replicates. This topsoil layer receives most plant input and gets most likely eroded and transported into the river. SPM (n=40) and riverbed sediments (n=37) were collected in a dry (February/March 2015) and wet (July/August 2015) season. For SPM, surface river water (10–80 L) was collected at mid-channel position from a bridge or boat, or 2–3 m out of the riverbank, and filtered on pre-combusted (450 °C, 6 h) GFF filters (0.7 μm, Whatman) using pressurized steel filtration units (after Galy et al., 2007). Additional river depth profiles (2–3 depths, 1–3 sites across river) were sampled in the Godavari delta and in the Middle Godavari (Fig. 1b; site 10 and 28). At these sites, river water was collected at equal increments to the riverbed with a custom-built depth sampler (after Lupker et al., 2011). Riverbed sediments were dredged at each location with a sediment grabber (Van Veen grab 04.30.01, Eijkelkamp) or with a shovel when the water level was low. The fine fraction (≤63 μm) was isolated by sieving for a selection of sites for soils (n=10) and riverbed sediments collected in the wet season (n=25). All samples were frozen upon arrival in the laboratory.

### 2.3. Elemental and bulk isotopic analysis

Prior to analysis, C3 and C4 plants, bulk soils and riverbed sediments were freeze-dried, homogenised and ground into powder using an agate mortar and pestle or a steel ball-mill. The bulk soils and sediments were decalcified by overnight treatment with 1 M HCL, then rinsed twice with deionised water and left to dry at 60

°C, following van Helmond et al. (2017). SPM was decalcified by vapour acidification (Komada et al., 2008; van der Voort et al., 2016). In short, randomly selected, small pieces of GFF filters containing the SPM were placed in pre-combusted (450°C, 6h) Ag capsules and put in a desiccator at 70 °C with 37 % HCl for 72 h and subsequently dried for minimal 120 h with NaOH. Fine fraction (≤63 μm) soils and sediments were placed in pre-combusted Ag capsules and decalcified by addition of 100 μL 1 M HCL and then left to dry overnight at 60 °C, following Vonk et al. (2008, 2010).

Total Organic Carbon (TOC) content (weight %) and bulk stable carbon isotopic composition ($\delta^{13}C$) of plants, (bulk) soils, riverbed sediments and SPM was measured with a Flash 2000 Organic Element Analyser connected to a Thermo Delta V Advantage isotope ratio mass spectrometer (Thermo Scientific, Italy), at NIOZ (Texel, The Netherlands). Total Nitrogen (TN) (weight %) was measured for plants and in non-decalcified bulk soils in the same way. Integration was performed with Isodat 3.0 software. TN in non-decalcified bulk sediments was measured with a NA 1500 NCS Analyser (Fisons Instruments, United Kingdom), at Utrecht University (Utrecht, The Netherlands). Fine fraction (≤63 μm) soils and riverbed sediments were analysed with a NC2500 Elemental Analyser coupled to a Thermo Finnigan DeltaPlus isotope ratio mass spectrometer (ThermoQuest, Germany), at VU University (Amsterdam, The Netherlands). The results were normalized to certified standards (Acetanilide, Benzoic acid and Urea at NIOZ and USGS40, USGS41 and IAEA601 at VU University), with an analytical uncertainty <0.1 % for TOC, <0.2 ‰ for $\delta^{13}C$ and <3 % for TN, based on replicate analysis of standards and samples. The $\delta^{13}C$ values were reported in the standard delta notation, relative to the international Vienna Pee Dee Belemnite (VPDB) standard for $\delta^{13}C$. C/N ratios were reported as mass ratios.

## 2.4. Precipitation and regression analysis

The Mean Annual Precipitation (MAP) in the Godavari basin was used to evaluate the control of drought stress on plant $\delta^{13}C$ values, as prior studies found evidence for a relationship between MAP and $\delta^{13}C$ values of C3 plants around the world (Stewart et al., 1995; Diefendorf et al., 2010; Kohn, 2010). Nonetheless, field surveys and data compilations of C3 vegetation in drought-stressed regions reported high inter- and intraspecies variation in C3 plant $\delta^{13}C$ values in response to MAP (Ma et al., 2012; Liu et al., 2013, 2014; Basu et al., 2021; Luo et al., 2021). The range of ~500 to 1500 mm y$^{-1}$ MAP in the Godavari basin was markedly lower than in tropical forests where MAP is typically >2000 mm y$^{-1}$ and where the majority of global C3 biomass occurs (Kohn, 2010). Here, we focused on MAP in 2014, the growing season preceding the sampling campaign in the

dry season in early 2015, considering that the majority (>80 %) of rainfall falls in the wet season and that dry to
moist deciduous vegetation is prevalent, which grows new leaves over the wet season and sheds them at the end
of the dry period (Kushwaha and Singh, 2005; Elliott et al., 2006). Long-term rainfall deficiencies have resulted
in pronounced drought conditions in the upper basin (Kirkels et al., 2021a) (Fig. S2).

In order to deal with the uneven distribution of analysed C3 plants over the MAP range in the Godavari basin,
we used a binning approach for the C3 plant $\delta^{13}$C values. The data were binned by calculating the average and
standard error of C3 plant $\delta^{13}$C values per MAP range of 100 mm y$^{-1}$. These binned Godavari C3 plant data were
subsequently plotted against the average MAP of each bin and utilised for regression analysis to assess the
relation between C3 plant $\delta^{13}$C values and MAP. The correlation established by regression analysis and the
(sub-)basin specific MAP was subsequently used to correct the plant stable carbon isotope end-member values
that are used in the C3/C4 mixing model for drought effects. A cut-off value for MAP of 1750 mm y$^{-1}$ was used
to determine plant end-members without drought effects, as above this MAP the C3 plant $\delta^{13}$C value can be
considered constant as there is no water limitation (Kohn, 2010).

### 2.5. Mixing model and Suess correction

The relative abundance of C3 and C4 plants was estimated based on the isotope mixing model by Philips and
Gregg (2001) using linear mass-balance equations and accounting for the variation in the C3 and C4 plants (i.e.,
sources) as well as in the soils or sediment (i.e., mixture):

$$\%C3 = [(\delta^{13}C_S - \delta^{13}C_{C4})/(\delta^{13}C_{C3} - \delta^{13}C_{C4})] * 100\% \qquad \text{Eq. (1)}$$

$$\%C4 = 100 - \%C3 \qquad \text{Eq. (2)}$$

In Eq. (1) $\delta^{13}C_{C3}$ and $\delta^{13}C_{C4}$ represent the bulk $\delta^{13}$C plant end-member values (‰) and $\delta^{13}C_S$ is the (sub)basin-
specific bulk $\delta^{13}$C value (‰) of soil or riverbed sediments. The $\delta^{13}C_S$ values were concentration-weighted using
the TOC content (weight %) of the individual samples in the (sub)basin and error propagation was accounted
for. This mixing model provided an estimation of the proportion of C3 and C4 plants, including the standard
error of variance on these estimates. Alternative mixing approaches including the C3 fraction woody cover,
which accounts for vegetation structure and shading effects (e.g., Wynn and Bird, 2008; Cerling et al., 2011;

Garcin et al., 2014), may be complicated by the fact that agricultural use (~60% of the basin) and deforestation since the 19[th] century have resulted in a more open landscape and has drastically reduced the area covered by native, closed-canopy forests, which is now limited to the East Tributary region.

C3 and C4 plant end-members to resolve the mixing model can be based on measurement of regionally occurring, modern vegetation in the Godavari basin (referred to as Godavari-based or regional end-members), which are representative of the prevailing habitat conditions. Alternatively, global end-members can be used based on C3 and C4 plants collected worldwide and reported in literature compilations. Commonly quoted global averages are -27 ‰ for C3 plants (Cerling et al., 1997; Koch, 1998; Dawson et al., 2002) and -12 ‰ for C4 plants (Koch, 1998; Dawson et al., 2002). However, the atmospheric $\delta^{13}C$ of $CO_2$ has rapidly declined over the past decades due to fossil fuel burning, so $\delta^{13}C$ data based on analyses of plants in the past requires a correction for this so-called Suess effect. Unfortunately, the exact sampling year for these global averages was unknown, but considering that similar values have been reported since the late 1970's we estimated a maximum decrease in global plant $\delta^{13}C$ values of ~0.9 ‰ (atmospheric $CO_2$ ~-7.5 ‰ in 1978 to ~-8.4 ‰ in 2015; Keeling et al., 2001, 2017; Graven et al., 2017; NOAA, accessed 21/6/2022). This correction for the Suess effect translated in to estimates of ~-27.9 ‰ and -12.9 ‰ for the global averages for C3 and C4 plants, respectively (Table 1). Alternatively, Kohn (2010) determined a value of ~-28.5 ‰ (corrected for Suess effect for the year 2000) representing global C3 vegetation including equatorial and mid-latitude biomass, which was updated to a modern value of ~-28.9 ‰. To compare $\delta^{13}C$ values measured in Godavari plants collected in early 2015 with those reported in earlier studies, we updated the latter to account for the Suess effect (Table 1).

**Table 1: Suess correction of plant $\delta^{13}C$ values**

| Type | Uncorrected $\delta^{13}C$ | Suess corrected $\delta^{13}C$ Modern (i.e., 2015)[1] | Suess corrected $\delta^{13}C$ Soil age (i.e., 1985)[1] |
|---|---|---|---|
| Global C3 vegetation | -27 | -27.9 | -27.1 |
| Global C3 vegetation (Kohn , 2010) | -28.5 | -28.9 | -28.1 |
| Global C4 vegetation | -12 | -12.9 | -12.1 |
| Measured C3 plants Godavari | -28.5 | —[2] | -27.7 |
| Measured C4 plants Godavari | -14 | —[2] | -13.2 |

[1] Based on Keeling et al. (2001, 2017) and Graven et al. (2017)

[2] Measured plants in Godavari basin are modern.

The plant $\delta^{13}C$ signal is subsequently transferred to soils or sedimentary deposits, where the $\delta^{13}C_{org}$ signal is assumed to integrate long-term and/or spatial areas and thus incorporate/average the plant $\delta^{13}C$ signal for a

range of precipitation within this period/region. In order to compare the $\delta^{13}C_{org}$ values of pre-aged soils and sediments with those of modern vegetation a correction for the Suess effect is warranted. Analysis of $\Delta^{14}C$ of OC in a selection of Godavari soils and sediments by Usman et al. (2018) revealed no distinct differences between the upper and lower basin nor between bulk soils and riverbed sediments. However, the large variation in $\Delta^{14}C_{OC}$ values, potentially related to small contributions of very old OC from wind-blown coal dust from the open-pit mines in the north of the basin, made it difficult to determine the average age of OC in Godavari basin based on these data. Assuming OC turnover rates of ~10 years in tropical forest soils to ~25 – 40 years in savanna ecosystems (Martin et al., 1990; Bird et al., 1996), we estimated an average age of ~30 years for OC in Godavari soils and riverbed sediments. This estimate is at the upper end of recently determined biome-specific OC turnover rates for tropical forests and savannas, where precipitation was shown to have a major effect on soil OC turnover rates (e.g., Carvalhais et al., 2014; Hein et al., 2020). To enable direct comparison of $\delta^{13}C$ values from plants with $\delta^{13}C_{org}$ values from soils and sediments and employ these in the mixing model, we corrected the measured $\delta^{13}C$ in modern vegetation for the Suess effect to the average age of soil/sediment OC (i.e., 30 years preceding the plant collection in 2015: 1985) (Table 1).

To evaluate the impact of C4 and C3 plant end-members on the reconstructed vegetation distribution, we compared four scenarios that included average global and regional vegetation end-members, and C3 plant end-members with and without a correction for a drought-induced $^{13}C$-enrichment (i.e., drought correction). This included: (1) Godavari C4 and C3 plant end-members, with drought correction of C3 plant $\delta^{13}C$ (2) Godavari C4 and C3 end-members, with no drought correction of C3 plant $\delta^{13}C$ (3) C4 and C3 end-members based on global data compilations, with the C3 end-member according to Kohn (2010) that includes equatorial and low-latitude biomass, and (4) global C4 and C3 end-members based on global data compilations and commonly quoted in literature. All plant end-members were Suess corrected to the equivalent age of soils and sediments in the Godavari basin (Table 1).

### 2.6. Statistics

Spatial and seasonal differences were evaluated with (Welch's) one and two-way ANOVA, (paired) t-tests and non-parametric Mann-Whitney and Kruskal-Wallis tests with R software package for statistical computing (R4.0.4; RStudio, v. 1.2.5033) and SPSS (IBM, v. 27.0.1.0). The level of significance was $p \leq 0.05$. The reported values are the mean ± standard error (SE). Linear regression analysis (Pearson's R) was performed to obtain the

correlation between $\delta^{13}C$ and MAP. Spatial patterns were further investigated with ArcGIS software (ESRI, v. 10.8.1).

## 3. Results and Discussion

### 3.1. Modern C3 and C4 plants in the Godavari basin and control by MAP

The Godavari plants (n=96) showed two distinct groups, with bulk $\delta^{13}C$ values that ranged from -12.7 to -15.1 ‰ for C4 plants (n=16, 9 different species) and from -24.3 to -33.2 ‰ for C3 plants (n=77, 38 different species) (Kirkels et al., 2021a) (Fig. 2 3a). The sampled Godavari plants fell within the typical ranges for global C4 (~– 10.5 to –14.5 ‰; Cerling et al., 1997) and C3 vegetation (~-20.5 to –37.5 ‰; Kohn, 2010) (downward corrected by ~-0.5‰ for fossil fuel burning). As in the Godavari basin only a very small area was covered by wet evergreen forest with a MAP of ~1500 – 2000 mm y$^{-1}$, we observed less negative $\delta^{13}C$ values for C3 plants compared to tropical rain forests with MAP typically exceeding 2000 mm y$^{-1}$ (Kohn, 2010) (Fig. 1b, 2).

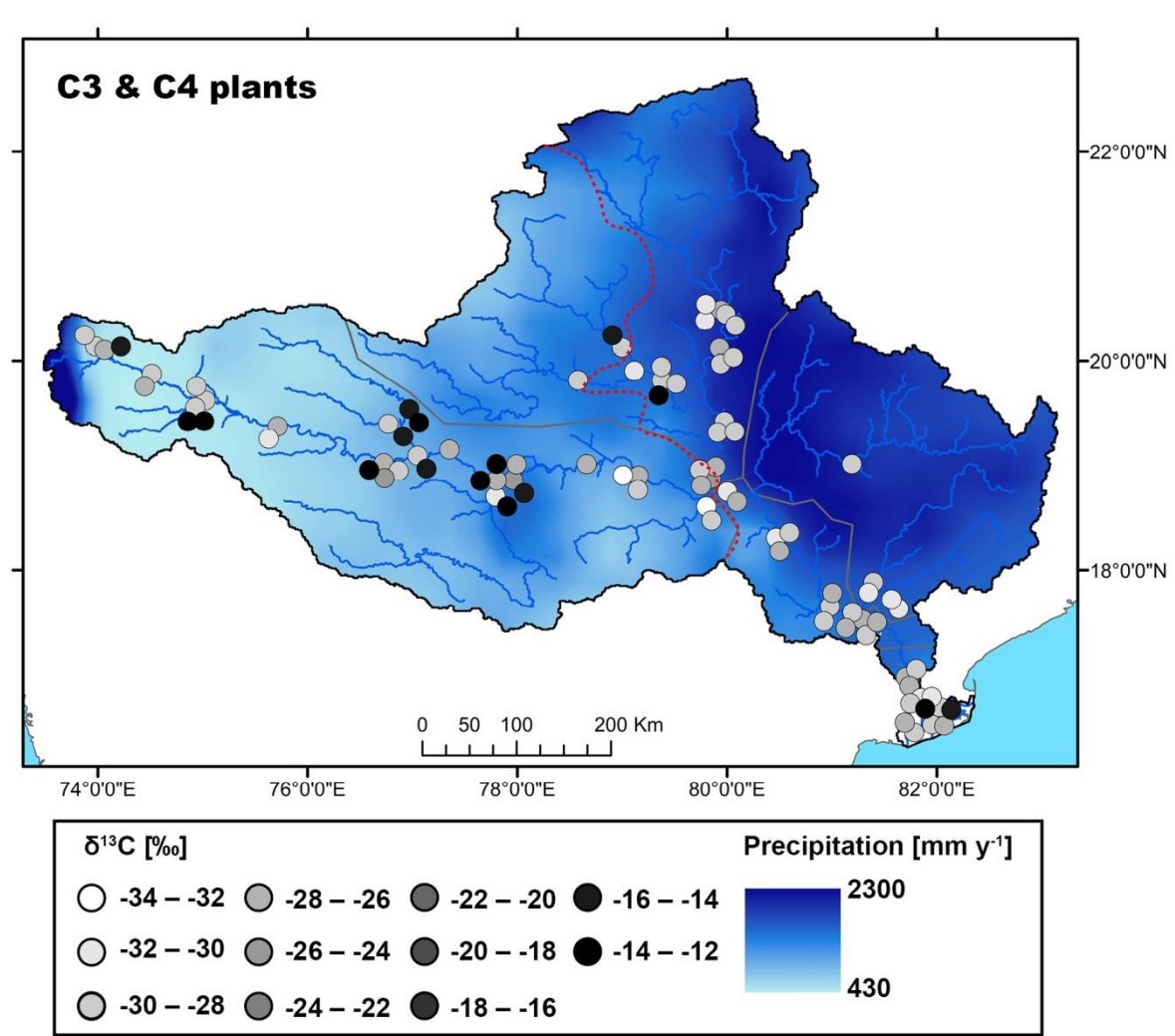

Fig. 2: Map showing the spatial distribution of C3 and C4 plant δ¹³C values in the Godavari basin. The red dashed line indicates the upper/lower basin boundary. The points refer to the measured δ¹³C values and the 30-year average rainfall distribution (MAP; 0.25°, APHRODITE dataset; Yatagai et al., 2009) is shown on the background.

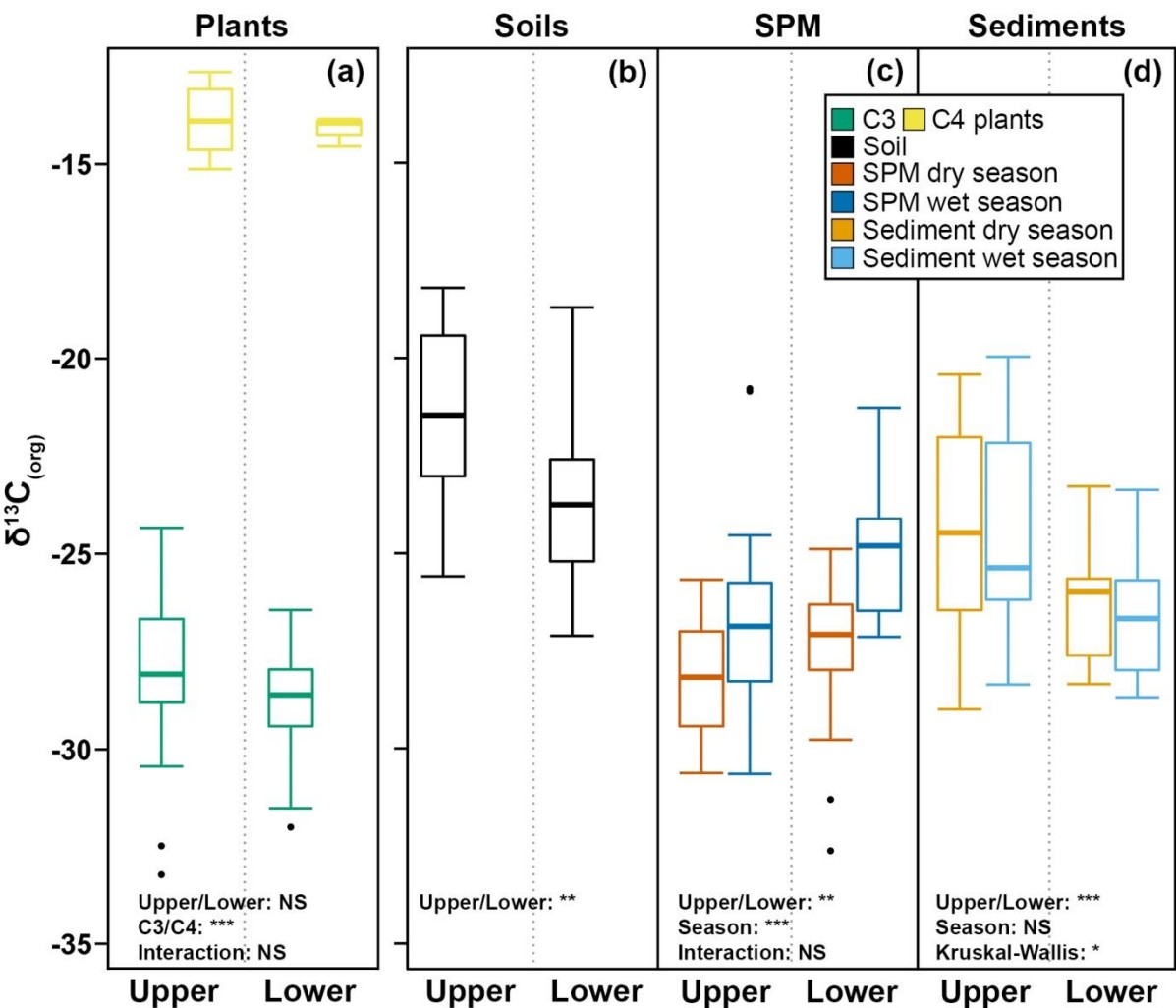

Fig. 3: Box-and-whisker plot of δ¹³C$_{(org)}$ values measured in C3 and C4 plants, soils, SPM and riverbed sediments collected in the dry and wet season in the upper and lower Godavari basin. The box represents the first (Q1) and third (Q3) quartiles, and the line in the box represents the median value, the whiskers extent to 1.5*(Q3–Q1) values and outliers are shown as points. Outcomes of the two-way ANOVA are indicated for plants, soils and SPM, and of non-parametric tests (Mann-Whitney and Kruskal-Wallis) for the sediments. The level of significance is: (NS) not significant, * $p \leq 0.05$, ** $p \leq 0.01$ and *** $p \leq 0.001$.

The C4 crops we collected in the Godavari basin, including *Z. mays, S. vulgare* and *S. officinarum* had similar $\delta^{13}$C values as earlier reported for these species in the Godavari basin (Pradhan et al., 2014; Krishna et al.,

2015). The Godavari C4 plants we sampled had on average more negative $\delta^{13}$C values than those collected in the only other extensive field survey of Indian plants on the Gangetic plain (-14.0±0.2 ‰ (±standard error: SE), n=16, (Fig. 3a) vs -12.7±0.2 ‰, n=45; p≤0.001), where they found most negative $\delta^{13}$C values in areas with MAP <1000 mm y$^{-1}$ and observed an effect of MAP on C4 plant $\delta^{13}$C values (Basu et al., 2015). In contrast, the Godavari C4 plants showed no significant correlation with MAP (Eq. 3; Pearson's R = -0.10; p=0.70) (Fig. 4).

The absence of a correlation for the Godavari C4 plants may be influenced by the relatively small sample size and their main occurrence in only a limited part of the MAP range covered (i.e., ~500 – 900 mm y$^{-1}$ in 2014). Nevertheless, earlier studies also predominantly found no trends in C4 plant $\delta^{13}$C values in response to MAP in dry ecosystems around the globe (<800 mm y$^{-1}$; Schulze et al., 1996; Swap et al., 2004). This finding was attributed to the $CO_2$ concentrating mechanism in C4 plants. This adaption to water loss due to evaporation in

warm and dry climates may be influenced by leaking of $CO_2$ from bundle sheath cells during extreme drought, but functions relatively robustly for a wide range of environmental conditions, including drought stress (Murphy and Bowman, 2009). Taken together, we interpret that there was no basis for correction of the $\delta^{13}$C C4 end-member for drought conditions.

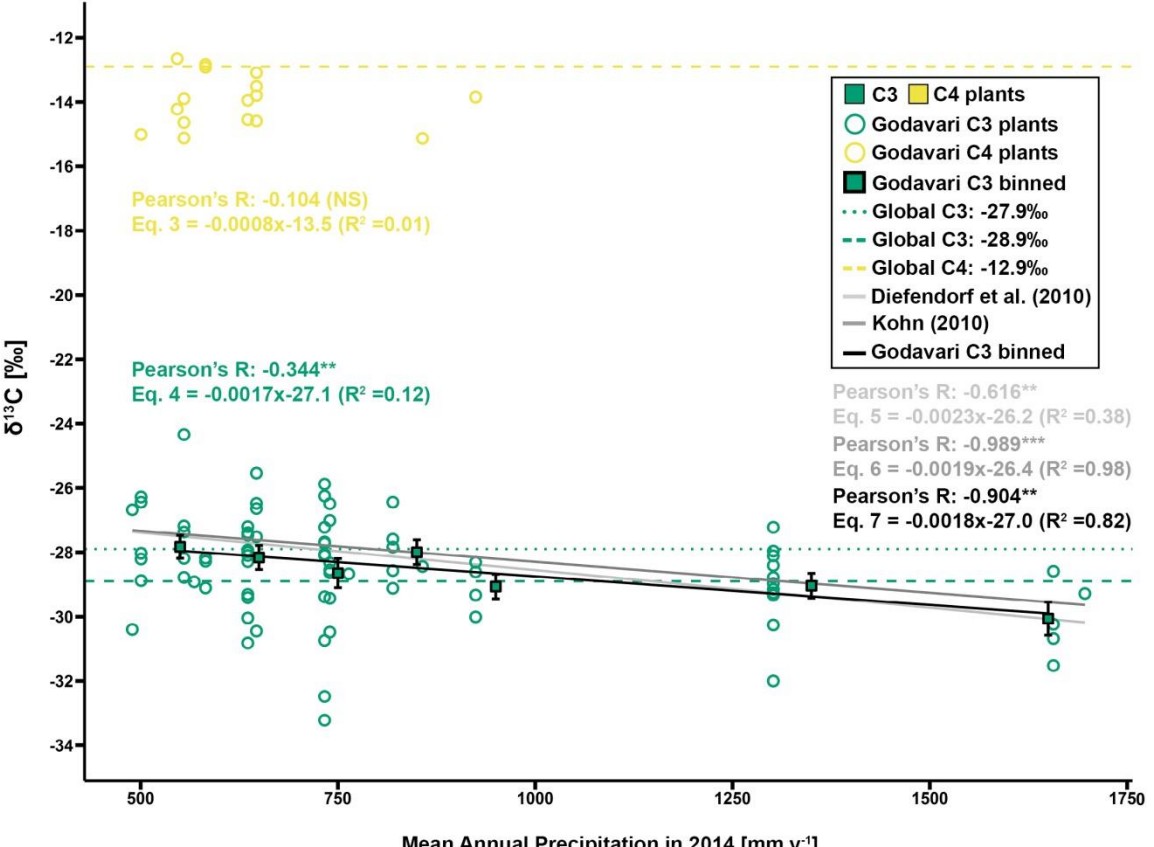

**Fig. 4: Regression analysis of $\delta^{13}C$ values against MAP (2014, previous growing season) for C3 and C4 plants in the Godavari basin and estimates based on global C3 vegetation models by Diefendorf et al. (2010) and Kohn (2010). For Godavari C3 plants, $\delta^{13}C$ values are also binned per MAP of 100 mm $y^{-1}$, and the mean ± standard error (SE; whiskers) is presented. The bin of 500–600 mm $y^{-1}$ includes two samples with a MAP of 489 mm. The solid lines denote the linear fit for the Diefendorf et al. (2010), Kohn (2010) and C3 binned correlation. The Pearson's R, equation and $R^2$ are given for each correlation. Compilation-based $\delta^{13}C$ values of vegetation sampled around the world are given for C4 (-12.9 ‰) (yellow dashed) and C3 plants (see Table 1). The latter is represented by a commonly quoted $\delta^{13}C$ value of -27.9 ‰ (dark green dotted) that is biased toward dry ecosystems according to Kohn (2010) and a $\delta^{13}C$ value established by Kohn (2010) of -28.9 ‰ (dark green dashed) that includes more equatorial and mid-latitude C3 biomass. The level of significance is: (NS) not significant, \* $p \leq 0.05$, \*\* $p \leq 0.01$ and \*\*\* $p \leq 0.001$.**

The average $\delta^{13}C$ value of Godavari C3 plants was significantly different in the upper and lower basin, where values are slightly less negative in the upper basin (-28.0±0.3 ‰, n=32) than in the lower basin (-28.8±0.2 ‰, n=45; $p \leq 0.05$) (Fig. 3a). This finding corresponds to the observed spatial gradient in MAP in the Godavari basin, and thus suggests an effect of MAP on C3 plant $\delta^{13}C$ values. Indeed, long-term MAP (1901–2015) was markedly lower in the upper than the lower Godavari basin ($p \leq 0.001$) and this contrast became more extreme for the 5-year average and 2014 MAP, which resulted in drought conditions in the upper basin (Kirkels et al., 2021a) (Fig. S2). The least negative subbasin-averaged $\delta^{13}C$ value for C3 plants was found in the Upper Godavari subbasin that received least precipitation ($\delta^{13}C$: -28.0±0.3 ‰, n=30; MAP: 593±18 mm $y^{-1}$) , compared to the most negative value in the East Tributaries that received significantly more precipitation ($\delta^{13}C$: -30.1±0.5 ‰, n=5; $p \leq 0.05$; MAP: 1530±142 mm $y^{-1}$; $p \leq 0.01$) (Fig. 2, S2). The East Tributaries are the only part of the Godavari basin that is covered by native, wet to moist forests where a denser canopy caused ample shading. This likely resulted in lower soil temperatures and higher moisture and humidity levels in the understory, which generally favours C3 vegetation (Cerling et al., 2011), that was indeed exclusively found in this Godavari subbasin (Fig. 1b, 2). Likewise, Garcin et al. (2014) reported very negative stable carbon isotopic values for dense tropical forests in Cameroon (>80% tree cover) mainly controlled by water availability, although a 'canopy effect' (van der Merwe and Medina, 1991), which involves recycling of $^{13}C$-depleted $CO_2$ in the understory of closed-canopy forests and stable carbon isotope fractionation due to photosynthesis under low light conditions, may have resulted in additional $^{13}C$-depletion in the C3 leaves.

The individual Godavari C3 plants revealed a small, but significant effect by MAP on their $\delta^{13}C$ values (Eq. 4; Pearson's R = -0.34; p≤0.0.1) (Fig. 4). This finding supports earlier studies by Diefendorf et al. (2010) and Kohn (2010) that found a strong control by MAP on C3 plant $\delta^{13}C$ on a global scale and established quantified relationships between the fractionation of carbon isotopes and environmental conditions, including MAP.

Application of these established relations for the Godavari basin revealed very similar trends as the Godavari C3 plants (Eq. 5 and Eq. 6, respectively; Fig. 4), although the Kohn (2010) relation was most similar in terms of slope and had a higher $R^2$ than the Diefendorf et al. (2010) correlation. For the individual Godavari C3 plants, we noted considerable variation in $\delta^{13}C$ values for any certain amount of precipitation, in line with earlier studies that found high inter- and intraspecies variation in C3 plant $\delta^{13}C$ values in response to MAP (Ma et al.,

2012; Liu et al., 2013, 2014; Basu et al., 2021; Luo et al., 2021). Moreover, the individual Godavari C3 plants were not evenly distributed over the entire precipitation range, making it more difficult to establish a correlation. Together, this resulted in a relatively weak linear correlation between MAP and individually measured C3 plants (Eq. 4; $R^2 = 0.12$), where MAP explained only ~12% of the variation in C3 plant $\delta^{13}C$ values.

Subsequent binning of C3 plant $\delta^{13}C$ values to overcome their uneven distribution over the range of MAP, revealed a strong and significant correlation with MAP (Eq. 7; Pearson's R = -0.90; p≤0.0.1) (Fig. 4). The slope of this binned C3 plant correlation (i.e., -0.18 ‰ per 100 mm MAP) could be used to estimate the offset of measured plant $\delta^{13}C$ values to those expected as a function of MAP. For the binned C3 plants, MAP explained ~82% of the variation in $\delta^{13}C$ values. This linear relation established here applies to the interval of ~500 to 1750

470   mm y$^{-1}$ precipitation, above which the C3 plant $\delta^{13}C$ value is assumed to be constant as there is no water limitation (Kohn, 2010). For very dry ecosystems with MAP <500 mm y$^{-1}$, non-linear effects on C3 plant $\delta^{13}C$ values need to be considered due to extreme drought stress (Kohn, 2010, 2011; Freeman et al., 2011). Similar to the Godavari C3 plants, a meta-analysis for low latitude regions (11–30°N) showed that the average C3 plant $\delta^{13}C$ value would change by ~-0.2 ‰ for every 100 mm increase in MAP for the interval of 500–1500 mm y$^{-1}$

(Basu et al., 2019b). We note that although the slope of the binned Godavari C3 plants was similar to that of the Kohn (2010) correlation, the ~0.6 ‰ offset in intercept suggests consistently more negative $\delta^{13}C$ values for Indian C3 vegetation in relation to MAP than established based on global data compilations by Kohn (2010) and Diefendorf et al. (2010). Within the MAP interval of ~1000–1500 mm y$^{-1}$, Godavari C3 plants had similar $\delta^{13}C$ values (-29.0±0.4 ‰, n=11; p>0.20) as those collected on the Gangetic plain situated in the Himalayan foreland

 (-29.6±0.2‰, n=76; Basu et al., 2015). The strong control by MAP on C3 plant $\delta^{13}$C values we established for the Godavari basin suggests that this factor needs to be considered in mixing model approaches to reconstruct vegetation in regions influenced by the Indian monsoon.

### 3.2. Tracing the plant $\delta^{13}$C signal along the plant–soil–river continuum

The bulk $\delta^{13}C_{org}$ signal preserved in soil or sedimentary archives depends on the input and integration of the of C3 and C4 plant-derived $\delta^{13}$C signal, where a temporal shift of ~0.8 ‰ due to the Suess effect is considered regarding the ~30 year turnover rate of OC in soils and sediments (Martin et al., 1990; Bird et al., 1996; Usman et al., 2018) (Table 1). Regardless, $\delta^{13}C_{org}$ may also be influenced by hydroclimatic controls on OC degradation and stabilisation mechanisms in soils and in the river (e.g., Carvalhais et al., 2014; Ward et al., 2017; Hein et al., 2020; Eglinton et al., 2021). The river-transported OC, in the form of suspended (SPM) or riverbed sediments, may be a complex mixture depending on soil- and plant/litter-derived OC sourcing from particular parts of the basin following rainfall distributions, soil mobilisation, aquatic primary production and hydrodynamic sorting processes within the river. This complexity warrants further exploration of the evolution and provenance of the $\delta^{13}C_{org}$ signal along the plant–soil–river continuum in the Godavari basin.

#### 3.2.1. Soils

The Godavari (bulk) soils had on average less negative $\delta^{13}C_{org}$ values in the upper than in the lower basin (-21.4±0.5 ‰, n=22 vs -23.5±0.5 ‰, n=25; p≤0.01) (Kirkels et al., 2021a) (Fig. 3b, 5b). This isotopic contrast corresponds with the vegetation distribution in the basin, with mixed C3 and C4 vegetation in the upper basin and more C3 plants in the lower basin (Fig. 1b, 2). The least negative $\delta^{13}C_{org}$ values were found in soils in the Upper Godavari and North Tributaries (-21.3±0.5 ‰, n=20 and -22.0±0.7 ‰, n=12, respectively) covered by thorny shrublands, dry deciduous forest and predominantly C4 crops, followed by the Middle Godavari (-23.5±0.5 ‰, n=4) in a transition zone, and most negative $\delta^{13}C_{org}$ values were found in soils in the East Tributaries and Lower Godavari that were covered by moist/evergreen forests and C3 crops (-24.7±0.6 ‰, n=6 and -25.1±0.6 ‰, n=5, respectively) (Fig. 1b, 5a,b, S3). These findings correspond with the general observation that the majority of the soil organic carbon derives from microbially processed plant residues, while the microbial biomass itself has been estimated to contribute only 1-5% (Kögel-Knabner, 2002; Simpson et al., 2007). We note that it is challenging to determine actual size of the microbial biomass, which is highly dependent on prevailing moisture levels, availability of easy degradable carbon as energy source and has a high

spatial heterogeneity (Birge et al., 2015; Wiesmeier et al., 2019). Given our sampling strategy where soils were collected in the dry season, the low moisture levels likely limited the microbial biomass size and activity in the Godavari soils. Hence, the $\delta^{13}C_{org}$ values in Godavari soils can be interpreted as a time-averaged plant signal on decadal scale that reflects the long-term hydrological conditions that underlie this vegetation distribution.

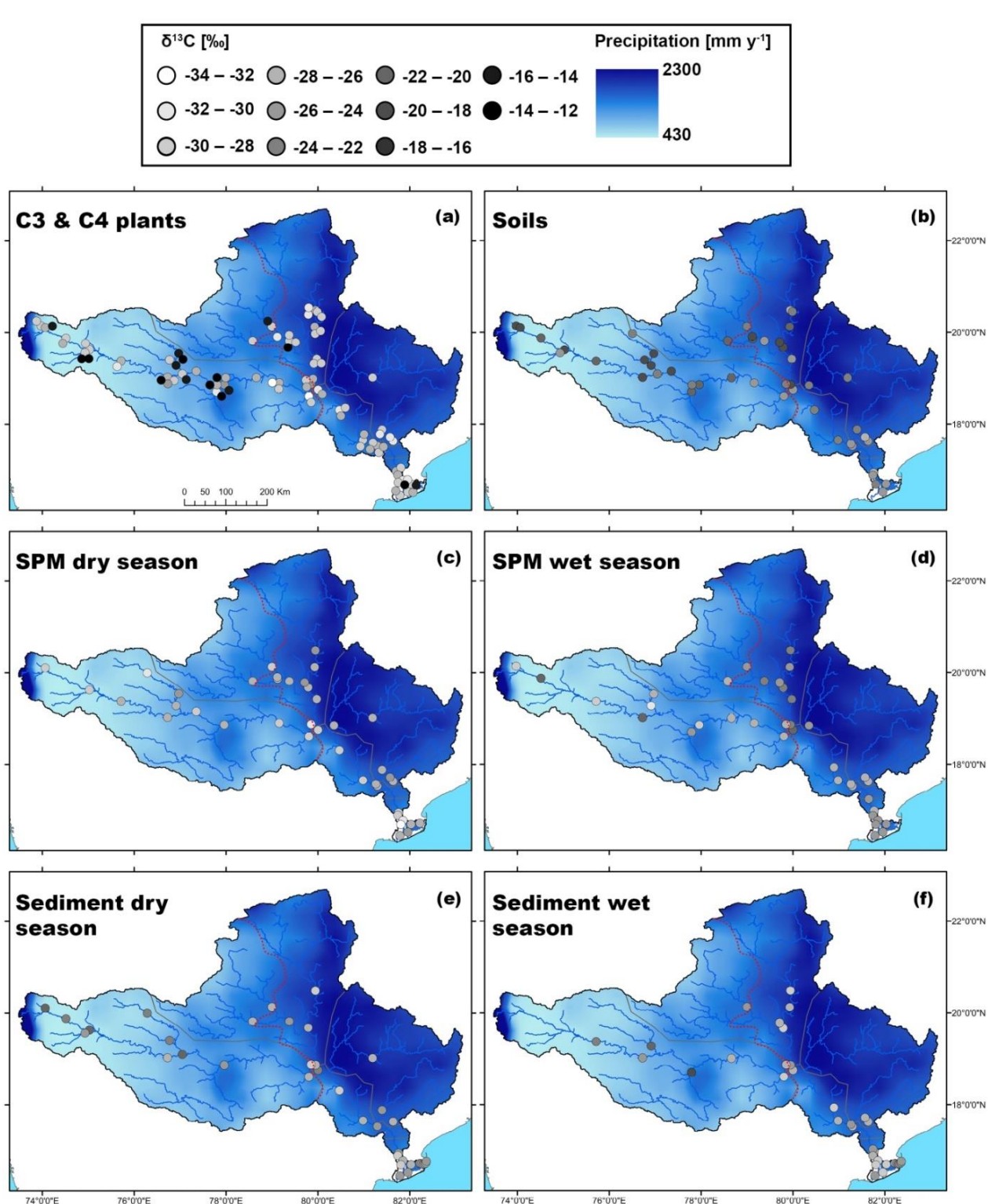

**Fig. 5: Maps showing the spatial distribution of (bulk) $\delta^{13}C_{(org)}$ values in the Godavari basin for (a) C3 and C4 plants, (b) (bulk) soils, SPM collected in the (c) dry and (d) wet season, and (bulk) riverbed sediments collected in the (e) dry and (f) wet season. The points refer to the measured $\delta^{13}C_{(org)}$ values and the 30-year average rainfall distribution is shown on the background.**

A potential degradation-related $^{13}C$-enrichment of the plant-derived OC in soils may be suggested by the ~4 ‰ difference between Suess-corrected C3 plant $\delta^{13}C$ and soil $\delta^{13}C_{org}$ values found in the C3-dominated lower basin (Fig. 3b, Table 1). This offset is relatively large compared to typical 1–3 ‰ higher $\delta^{13}C_{org}$ values due to soil OC degradation (e.g., Krull et al., 2005; Sreemany and Bera, 2020), which may result from preferential mineralisation of chemical compounds containing less $^{13}C$ (i.e., with a lower $\delta^{13}C$ value) and/or stable carbon isotope fractionation during microbial processing preferentially utilising $^{12}C$ over $^{13}C$ (Schmidt and Gleixner, 1998). Instead of a degradation-induced shift, deforestation since the late 19[th] century and agricultural expansion with predominantly drought-adapted C4 crops (Ponton et al., 2012; CWC, 2014; Pradhan et al., 2014) may have contributed to relatively more C4 input and thus enrichment of topsoil $\delta^{13}C_{org}$ values. Regardless, soil OC degradation and stabilisation processes in tropical to subtropical biomes may have an opposite effect on soil $\delta^{13}C_{org}$ signals. In mixed C3/C4 ecosystems, C4 plant-derived OC has been shown to contain more labile compounds and thus degrade more rapidly than C3 plant-derived OC that contains more difficult to degrade compounds (Wynn, 2007; Wynn and Bird, 2007). However, C4-derived OC has also been shown to be preferentially incorporated into fine fractions as fine particles are presumed to have a higher ability to stabilise the labile, C4-derived compounds onto mineral surfaces where they are better protected against degradation, whereas C3-derived OC is preferentially added to the coarse fraction thus leaving it less protected (Bird and Pousai, 1997; Wynn, 2007; Wynn and Bird, 2007). In the Godavari basin, Usman et al. (2018) reported similar $\Delta^{14}C$ values for soil OC in the upper and lower basin that have different C3 and C4 plant covers, suggesting that the nett effect of preferential degradation (more young OC) and stabilisation (more old OC) is minor. Indeed, extensive degradation of C4 plant-derived OC is unlikely, given that the upper basin with most C4 plants contains clay-rich, fine particles from weathering of the Deccan basalts (Giosan et al., 2017; Usman et al., 2018; Kirkels et al., 2021b), which would contribute to stabilise the C4-derived OC. Fine soils (≤63 μm) sampled in the upper basin had on average slightly less negative $\delta^{13}C_{org}$ values than the bulk soils (-19.7±1.0 ‰, n=8 vs -21.1±0.7 ‰, n=8; p≤0.09), pointing towards preferential stabilisation of C4-derived OC in the fine fraction (Fig.

S4). This result highlights that differences between bulk and fine size fractions could be important for the $\delta^{13}C_{org}$ signal preserved in soils.

### 3.2.2. Suspended particulate matter

For SPM collected in the Godavari River, bulk $\delta^{13}C_{org}$ values were consistently more negative in the dry than in the wet season, and more negative in the upper than the lower basin, but there was no significant interaction between the seasonal and upper/lower basin effects (Fig. 3c). In the dry season, SPM $\delta^{13}C_{org}$ values were significantly more negative than in Godavari soils (-27.8±0.3 ‰, n=40 vs -22.5±0.4 ‰, n=47; p≤0.001), making predominant soil–to–river input in this season unlikely (Fig. 3b,c, 5b,c). Instead, the quiescent waters behind dams and very low river discharge favour aquatic primary production (Pradhan et al., 2014). Freshwater phytoplanktonic matter usually has relatively low $\delta^{13}C_{org}$ values. The isotopic fractionation between phytoplankton and dissolved inorganic carbon (DIC) has been estimated at ~-23 ‰, resulting in typical phytoplankton-derived $\delta^{13}C_{org}$ values between -31 and -35 ‰ in the Ganges-Brahmaputra as well as at the start of the dry season in the Godavari River (e.g., Aucour et al., 2006; Galy et al., 2008b; Krishna et al., 2015). The observation of slightly less negative $\delta^{13}C_{org}$ values in our SPM collected at the end of the dry season may be explained by eutrophic conditions due to agricultural/wastewater inputs which fuelled intense aquatic production in the Godavari River (Balakrishna and Probst, 2005; Pradhan et al., 2014). During periods of high aquatic productivity, the carbon isotope fractionation becomes smaller (up to 0‰; Torres et al., 2012) and more $^{13}C$ gets incorporated into the phytoplanktonic biomass. Aquatic primary production is also supported by the strong increase in %OC from soils to dry season SPM (0.8±0.1 %, n=47 vs 11.4±1.1 %, n=39; p≤0.001), since phytoplankton-derived SPM is typically high in %OC (Aucour et al., 2006; Galy et al., 2008b). Notably, $\delta^{13}C_{org}$ values of dry season SPM became less negative near the Godavari's outflows into the Bay of Bengal (Fig. 5c, S3), suggesting mixing of freshwater and estuarine/marine phytoplankton in the delta, where the latter has typically less negative $\delta^{13}C_{org}$ values (i.e., -22.8 to -24.4 ‰; Dehairs et al., 2000; Krishna et al., 2015; Gawade et al., 2018). This observation is consistent with changes in electrical conductivity and water isotopic values ($\delta^{18}O$) that showed seawater intrusion in the delta in the dry season (Kirkels et al., 2020b). Regardless, mixing of riverine and marine OC with different stable carbon isotopic values at the outflow complicates the tracing of the Godavari-derived OC signal from the river mouth to marine sedimentary deposits.

In the wet season, there was a strong isotopic contrast between SPM collected in the upper and lower basin (-26.4±0.8 ‰, n=14 vs -25.0±0.3 ‰, n=26; p≤0.001) (Fig. 3c). The negative $\delta^{13}C_{org}$ values in the upper basin suggest continuous aquatic production, allowed by the limited rainfall and abundant dams in this region that created standing waters and facilitated year-round aquatic productivity (Pradhan et al., 2014; Kirkels et al., 2020b). A few sites in the upper basin had remarkably less negative $\delta^{13}C_{org}$ values, suggesting that some local soil or C4 plant input occurred at locations where agricultural fields with exposed topsoils were situated next to the river (Fig. 5d). In the lower basin, wet season SPM $\delta^{13}C_{org}$ values varied from -21.3 to -27.1 ‰ and fell within the range of bulk soils in this region (-18.7 to -27.1 ‰). This resemblance suggests contribution of soil-derived OC to wet season SPM in the lower basin. Water isotopic values and rainfall distributions confirmed substantial discharge in the wet season and identified the Weiganga/Pranhita rivers in the North Tributaries and the Indravati River in the East Tributaries subbasin as major source areas (Kirkels et al., 2020b). It is generally assumed that the high flow velocity and turbidity in the wet season would limit aquatic production as well as OC degradation during fluvial transport (Balakrishna and Probst, 2005; Acharyya et al., 2012). In contrast, Galy et al. (2008b) showed for the Ganges-Brahmaputra that ~50 % of the wet season SPM derived from the upper reaches of the basin was degraded during river transit and replaced by local input from the plains. Quantifying the extent of OC degradation during fluvial transport is thus not straightforward as it depends on the stability of OC in the river e.g., protected by mineral-associations or not and on its residence time in a specific basin (e.g., Ward et al., 2017; Eglinton et al., 2021).

### 3.2.3. Riverbed sediments

For riverbed sediments collected in the Godavari River, bulk $\delta^{13}C_{org}$ values were consistently less negative in the upper than in the lower basin (-24.4±0.6 ‰, n=21 vs -26.6±0.2 ‰, n=53; p≤0.001), but there was no significant seasonal effect (Fig. 3d, 5e,f). The latter suggests that riverbed sediments represented a season-integrated $\delta^{13}C_{org}$ signal, versus SPM that showed more seasonal variation. A Kruskal-Wallis test revealed that sediments collected in the upper basin in the dry season and those collected in the lower basin in the wet season were significantly different (-24.4±0.7 ‰, n=13 vs -26.6±0.3 ‰, n=29; (Bonferroni-corrected) p≤0.05) (Fig. 3d). However, riverbed sediments showed a high spatial variation in $\delta^{13}C_{org}$ values, which ranged from -20.0 to -29.0 ‰ in the upper and -23.3 to -28.7 ‰ in the lower basin (Fig. 5e,f). The strong control by upper/lower basin location on sediment $\delta^{13}C_{org}$ values corresponds to the vegetation distribution with more C4 plants and drought-stressed C3 plants in the upper basin, leading to less negative plant $\delta^{13}C$ and soil $\delta^{13}C_{org}$ values to be transferred

to the riverbed sediments. Riverbed sediment $\delta^{13}C_{org}$ values were consistently more negative than soil $\delta^{13}C_{org}$ values in the upper and lower basin (Fig. 3b,d), suggesting input from an additional, [13]C-depleted source, likely phytoplankton- or C3 plant-derived OC.

Possible contributions of additional sources can be further explored using a source diagram, where the relation between $\delta^{13}C_{org}$ values and C/N ratios reveals source-specific distributions which help to identify the provenance of OC sources in aquatic ecosystems (e.g., Lamb et al., 2006). For the Godavari basin, riverbed sediments plotted close to Godavari C3 plants, albeit at lower C/N ratios (Fig. 6). This may suggest that slightly degraded C3 plant-derived OC is selectively transported by/stored in the lower basin sediments. The lower C/N ratios suggest slight degradation of this OC, although there could also be a small contribution of phytoplankton-derived OC that settled onto the riverbed. Notably, the upper basin sediments plotted generally closer to the upper basin soils, suggesting an inherited soil $\delta^{13}C_{org}$ signal in these sediments, as soil input was diminished by limited rainfall in this region during our sampling campaigns.

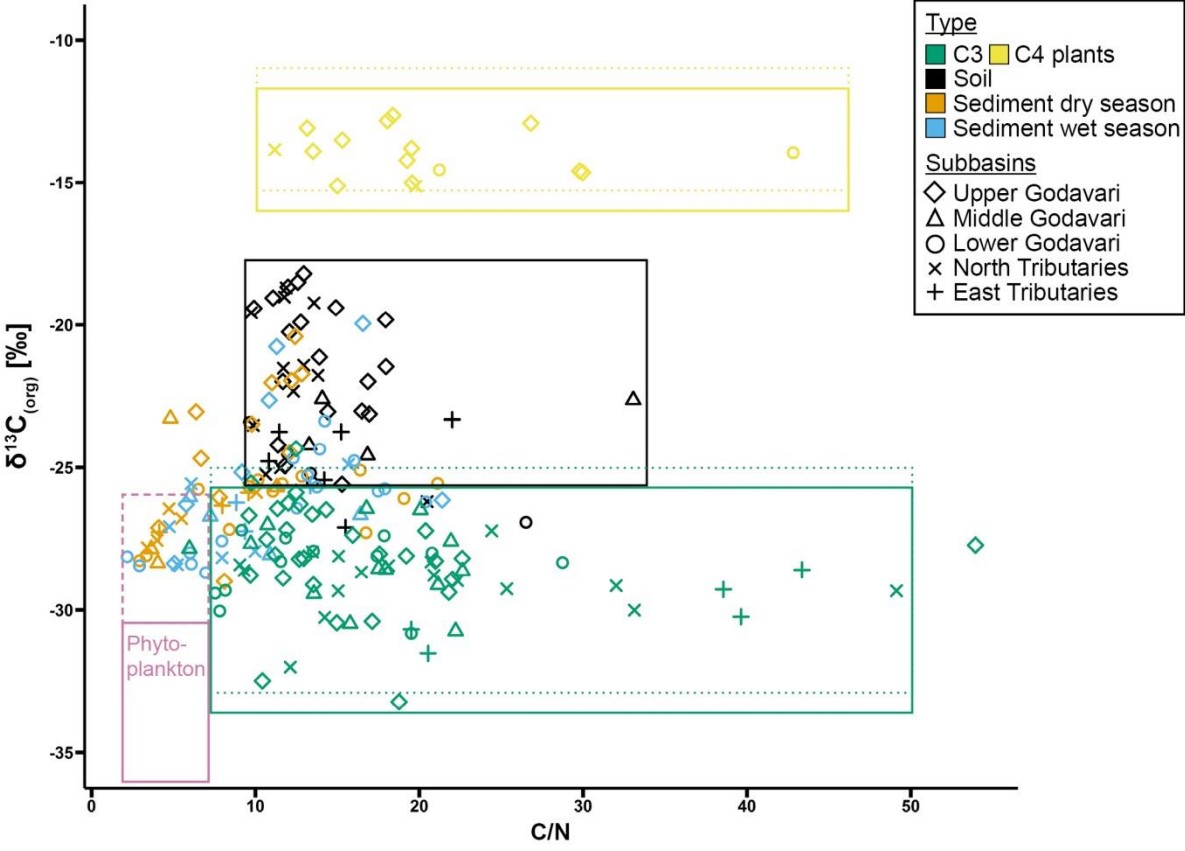

**Fig. 6: Source plot with relation between $\delta^{13}C_{org}$ values and C/N mass ratios for modern C3 and C4 plants and (bulk) soils and riverbed sediments collected in the dry and wet season in the Godavari basin. C/N ratios are retrieved from Kirkels et al. (2021c). Freshwater phytoplankton is defined for Indian**

 **monsoonal rivers in the dry season (solid line) (C/N: 2.5–8; data from Balakrishna and Probst, 2005;**

**$\delta^{13}C_{org}$: -31 – -35 ‰; data from Aucour et al., 2006; Galy et al., 2008b; Krishna et al., 2015), and based on**

**SPM collected at the end of the dry season in this study (dashed pink line). For C3 and C4 plants, Suess**

**corrected $\delta^{13}C$ values (equivalent to the age of soils and sediments, see Table 1) are indicated by the**

**dotted lines.**

### 3.2.4. Hydrodynamic sorting

Fluvial transport of fine particles is of particular interest as they are generally enriched in OC and their size facilitates effective offshore transport to sedimentary deposits (e.g. Bianchi et al., 2018). At the same time, coarse- and fine-grained particles have been shown to be sensitive to hydrodynamic sorting effects within the

 river, resulting in depth-specific OC distributions and/or preferential transport of certain OC components (Galy et al., 2008b; Bouchez et al., 2014; Feng et al., 2016; Repasch et al., 2022). Similar to the Godavari soils, riverbed sediments collected in the wet season revealed distinctly less negative $\delta^{13}C_{org}$ values for the fine fractions than the corresponding bulk sediments (-24.4±0.3 ‰, n=25 vs -25.7±0.4 ‰, n=25; p≤0.001) (Fig. 5f, S4). This difference corresponds to previous findings that C4 plant-derived OC is preferentially associated with

 finer fractions whereas C3 plant-derived OC is contained in the coarser fraction of soils (Bird and Pousai, 1997; Wynn and Bird, 2007) and river(-dominated) sediments (e.g., France-Lanord and Derry, 1994; Bianchi et al., 2002). This finding suggests differential transport of C3- and C4-derived OC in the fine and bulk Godavari riverbed sediments.

 Furthermore, wet season SPM collected along river depth profiles in the Middle and Lower Godavari showed in general more negative $\delta^{13}C_{org}$ values with depth (Fig. S5). Also, the riverbed sediments dredged at each location had more negative $\delta^{13}C_{org}$ values than the SPM in the water column above. This trend in $\delta^{13}C_{org}$ values suggests that C3-derived OC with typically more negative $\delta^{13}C_{org}$ values was transported in coarse-grained sediments and SPM near the riverbed, similar to findings at peak discharge for the Himalayan-derived Ganges-Brahmaputra

 (Galy et al., 2008b) and the Rio Bermejo draining the central Andes (Repasch et al., 2022). Alternatively, a [13]C-depleted phytoplankton source was unlikely given that the high flow velocity and turbidity in the monsoon season prevent light penetration and generally limit algae production (Balakrishna and Probst, 2005; Acharyya et al., 2012). Indeed, analysis of lignin, a macromolecule that is exclusively produced by plants, revealed a C3-derived 'woody undercurrent' in Godavari riverbed sediments collected in the wet season in the lower basin

(Pradhan et al., 2014), similar to earlier findings in the Madre de Dios and Mississippi rivers with strong seasonal variability in their hydrology (Bianchi et al., 2002; Feng et al., 2016). This information corroborates differential transport of fine and coarse-grained particles and their associated OC sources in the Godavari basin. This insight has important implications for the $\delta^{13}C_{org}$ signal that is finally exported to marine sedimentary deposits in front of the Godavari's mouth, where C4-derived OC with less negative $\delta^{13}C_{org}$ values may be

overrepresented, as fine fractions are transported farther offshore and in greater quantities than coarse-grained particles (Goñi et al., 1997, 1998; Bianchi et al., 2002). Indeed, Holocene marine sediments collected in front of the Godavari's mouth contained no woody particles (Ponton et al., 2012; Giosan et al., 2017; Usman et al., 2018), in contrast to the Bay of Bengal Fan fed by the Himalayan-derived Ganges-Brahmaputra, that covers a steep altitudinal gradient and carries coarse sediments far offshore at high flow conditions, where wood particles

were found in sediments spanning the last 19 Ma (Lee et al., 2019).

### 3.3. C3/C4 vegetation reconstruction and end-member analysis

Mixing model estimates based on the $\delta^{13}C_{org}$ signal in Godavari soils revealed a high %C4 plants in the Upper Godavari subbasin (43±3 %; scenario 1) and in general in the upper basin (43±3 %), that decreased toward the

lower basin (31±3 %), from the North Tributaries (40±5 %), Middle Godavari (32±4%) and East Tributaries (29±4 %) to the lowest %C4 plants in the Lower Godavari subbasin (19±4%) (Fig. 7a). This result is consistent with the vegetation structure in the Godavari basin, with a mixed C3/C4 vegetation in the dry upper basin and dominant C3 vegetation in the lower basin (Fig. 1b, 2) (Giosan et al., 2017; Usman et al., 2018). Pollen assemblages of the modern vegetation corroborate these findings, as they found ~50% *Poaceae* (here C4

grasses) in the upper basin based on surface sediments of the Lonar crater lake (Prasad et al., 2014; Riedel et al., 2015), and ~30 % *Poaceae* for the whole Godavari basin based on marine surface sediments in front of the Godavari mouth in the Bay of Bengal (Zorzi et al., 2015), which was close to our estimate of 37±2 % C4 plants for the whole basin. Notably, the estimated %C4 plants in the C3-dominated lower basin was relatively high (31±3 %), and may have been influenced by degradation-related enrichment of the soil $\delta^{13}C_{org}$ signal leading to

a potential overestimation of the %C4 plants. The C3/C4 abundances based on the $\delta^{13}C_{org}$ signal of riverbed sediments collected in the wet season also reflected the vegetation gradient (Fig. 7b, 1b). The estimated %C4 plants was consistently lower for riverbed sediments than for soils (-1 to -21 %; scenario 1), particularly in the North Tributaries (-19 %) and the Middle Godavari (-21 %) subbasins. Hydrodynamic sorting and preferential

transport of C3 plant-derived OC in the coarse-grained riverbed sediments may have influenced the C3/C4 plant

estimates, resulting in a potential underestimation of the %C4 plants in these basins.

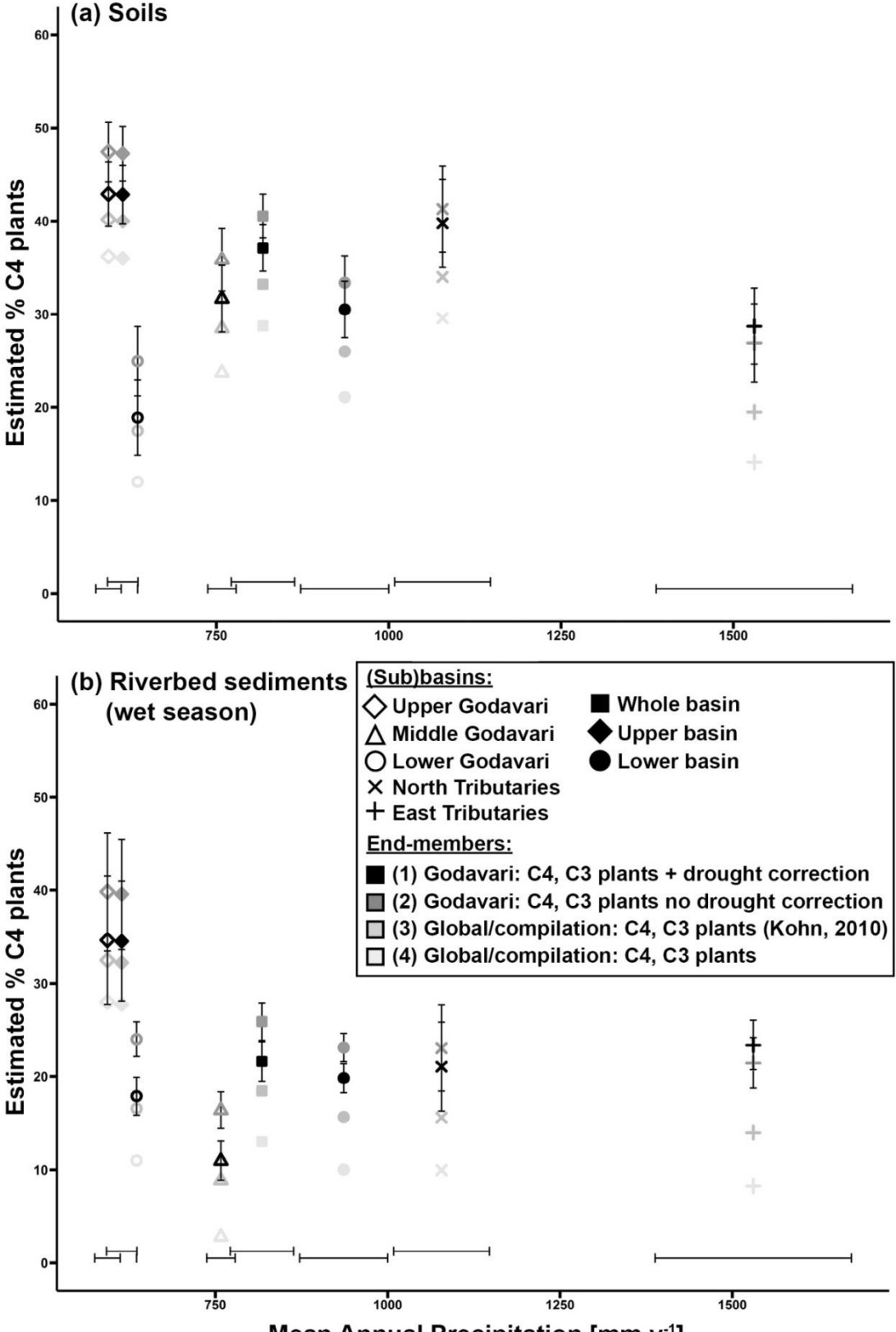

**Fig. 7: The estimated proportion (%) of C4 plants determined by a linear mixing model (by Philips and Gregg, 2001) versus MAP for different (sub)basins, based the OC-weighted $\delta^{13}C_{org}$ signal in (a) soils and (b) riverbed sediments collected in the wet season. The whiskers represent ± standard error (SE). Four scenarios are included: (1) Godavari C4 and C3 plant end-members, with drought correction of C3 plant $\delta^{13}$C (2) Godavari C4 and C3 end-members, with no drought correction of C3 plant $\delta^{13}$C (3) C4 and C3 (based on Kohn, 2010) plant end-members based on compilations of global vegetation, and (4) C4 and C3 plant end-members as commonly quoted in literature (e.g., Cerling et al., 1997; Koch, 1998; Dawson et al., 2002) based on global vegetation (see Table 1). For scenario (2) the mean $\delta^{13}$C value for Godavari C3 plants with no drought stress (i.e., -29.3 ‰) was determined at MAP of 1750 mm $y^{-1}$ using Eq. 7 (see Fig. 4).**

In order to assess the influence of drought stress and the use of region-specific plant $\delta^{13}$C end-members on C3/C4 vegetation mixing model estimates, we compared four scenarios (Fig. 7). Comparing end-members with and without drought correction of the Godavari C3 plant $\delta^{13}$C end-member (scenario 1 and 2, respectively) revealed a differential effect of MAP on the estimated %C4 plants. For MAP above 1500 mm $y^{-1}$ when water was probably no limiting factor for C3 plants (Kohn, 2010), the estimated %C4 plants was ~2 % lower without correction. But for MAP below 1500 mm $y^{-1}$, the estimated %C4 plants was 2–6 % higher without this correction for MAP, suggesting a potential overestimation of the %C4 plants. Notably, drought correction of the C3 plant $\delta^{13}$C end-member is increasingly important with decreasing MAP.

Comparing mixing model results for Godavari-specific end-members with drought correction and global average end-members for C4 plants (-12.1‰) and C3 plants (-28.1 ‰; based on Kohn, 2010) (scenario 1 and 3, respectively) revealed that the estimated %C4 plants was 1–9 % lower for the latter, suggesting a potential, minor underestimation. The estimated %C4 plants was 5–9 % lower for the wetter North and East Tributaries, while the difference was minor (≤5 % lower) for the other subbasins as well as for the whole, upper and lower basin. Notably, comparing Godavari-specific end-members with drought correction and commonly quoted global end-members for C4 (-12.1 ‰) and C3 plants (–27.1 ‰) (scenario 1 and 4, respectively), showed that the estimated %C4 plants was 7–15 % lower for the latter, suggesting a substantial underestimation when using these global end-members. This would result in a %C4 plants of only 36±3 % for the upper basin, where C4 grasses and crops are most abundant (Pradhan et al., 2014; Prasad et al., 2014; Riedel et al., 2015; Giosan et al.,

2017) and only 21±3 % C4 plants in the C3-dominated lower basin. Finally, the underestimation of %C4 plants was even larger (11–14 %) for all (sub)basins when comparing the latter global end-members with Godavari-specific end-members without drought correction (scenario 4 and 2, respectively). Our results from the Godavari basin highlight how important it is to make informed choices about the plant end-members used for vegetation reconstructions, considering characteristics of the regional vegetation and environmental controls such as MAP in regions where moisture limiting conditions may prevail (on a seasonal scale).

### 4.   Conclusion – Implications for vegetation reconstructions

Our analysis of contemporary C3 and C4 plants, soils and sediments from the Godavari basin has resulted in three important considerations for the reconstruction of C3/C4 vegetation distributions using bulk $\delta^{13}$C analysis. Firstly, our C3 and C4 plant $\delta^{13}$C data for the Godavari revealed the importance of making informed choices about the plant end-members used for vegetation reconstructions, considering the regional conditions including vegetation species, structure, land-use and environmental controls. In particular in tropical to subtropical regions with generally high vegetation biomass and more negative $\delta^{13}$C C3 plant values, the use of end-members representing these negative C3 plant $\delta^{13}$C values would result in a higher estimation of %C4 plants when applied to areas with less precipitation (MAP <2000 mm), such as the Godavari basin. Given that extensive datasets of the stable carbon isotopic values of commonly occurring plants in certain areas are not generally available, an alternative approach is to derive carbon stable isotope plant end-member values from data compilations of global vegetation, with consideration for the conditions prevailing in the area of interest. For the Godavari basin, using plant end-members based on measurement of Godavari plants resulted in a ~1–9 % higher estimated %C4 plants compared to using a compilation-based end-member value from Kohn (2010) that was considered representative of the vegetation in this region (Table 2). Correction for the Suess effect to account for fossil fuel burning over the last decades had a relatively minor impact (generally < 2%, depending on the plant end-member chosen) on the estimated %C4 plants (Table 2). The offset we observed for the Godavari basin thus can thus be attributed to the choice of plant end-members used in the mixing model.

Secondly, it is well-known that C3 plant $\delta^{13}$C values become less negative with decreasing amounts of precipitation, which may result in an underestimation of the %C4 plants. Incorporating a drought correction for the C3 plant end-member resulted in maximal 6 % lower %C4 plants for the different Godavari subbasins, but the extent of this effect will be most pronounced in C3-dominated ecosystems with low MAP. For the Godavari

basin, we established that the impact of drought correction on the estimated %C4 plants differed depending on the amount of MAP (Table 2). The effect was relatively minor (~2% lower %) for C3-dominated areas that received >1500 mm y$^{-1}$ and where water was no limiting factor, but became positive (~2% higher) for areas with 1000–1500 mm y$^{-1}$ rainfall and even larger (4–6 %) for areas with a MAP of ~500–1000 mm y$^{-1}$ and mixed C3 and C4 vegetation.

**Table 2: Summary of processes/factors influencing the C3/C4 mixing model estimates**

| Mixing model parameters | Compared to | Estimated %C4 plants |
|---|---|---|
| Regional plant end-members | Global end-members (C3 by Kohn, 2010) | ≈ / ↑[1] |
| | Global end-members | ↑ |
| Suess correction[2] | Modern C4 plants | ↓ |
| | Modern C3 plants | ≈ / ↓[3] |
| Drought correction | Godavari C4 plants, no drought correction | – |
|     MAP > 1500 mm y$^{-1}$ | Godavari C3 plants, no drought correction | ↑ |
|     MAP < 1500 mm y$^{-1}$ | Godavari C3 plants, no drought correction | ↓ |
| **Soil-river continuum** | **Specification** | |
| Soil degradation | More degradation C3-derived OC | ↑ |
| | More degradation C4-derived OC | ↓ |
| In-river OC processing | More phytoplankton production | ↓ |
| | More riverine OC degradation | ↑ |
| Hydrodynamic sorting | More coarse particles | ↓ |
| | More fine particles | ↑ |

[1] With / without drought correction of the regional end-member

[2] Equivalent to the OC turnover in soils and sediments (i.e., 1985)

[3] C3 plant (global) end-member of -27.1‰ / Godavari-based or -28.1‰ (global) end-member

Thirdly, C3/C4 vegetation reconstructions based on the $\delta^{13}C_{org}$ signal preserved in soils, riverine and/or marine sediments should take potential alterations in this signal along the plant–soil–river continuum into account. Differential degradation and stabilisation of C3- and C4-derived OC in soils and sediments, as well as phytoplankton contributions in the river and hydrodynamic sorting effects that separate coarse and fine fractions and their associated OC may result in a complex interplay affecting the mixing model estimates (Table 2). For the Godavari basin, our results suggest that the $\delta^{13}C$ values of C3 and C4 plants are relatively well-maintained throughout the transport chain, so $\delta^{13}C_{org}$ values of soils and riverbed sediments can be used as proxy for C3/C4 vegetation distributions.

Finally, our results in the Godavari basin highlight that precipitation plays an impactful role in the reconstruction of vegetation by influencing plant $\delta^{13}C$ end-members, OC decomposition rates and driving OC transport by rivers. Our results thereby support the assumption that in monsoon-influenced regions, wetter conditions/periods require no drought correction of the C3 plant end-member and generate more erosion and

rapid transport downstream, which generally limits degradation during river transit. This implies that the $\delta^{13}C_{org}$ signal exported to sedimentary deposits would reflect the C3/C4 vegetation distribution in the basin.

**Data availability**

Research data associated with this article are available in the open access Pangaea Data Repository (Kirkels et al., 2022; accessible via: https://doi.pangaea.de/10.1594/PANGAEA.940189). Background geochemical data on the Godavari basin and TN data (Kirkels et al., 2021; accessible via: https://doi.pangaea.de/10.1594/PANGAEA.937965) are also available in the Pangaea Data Repository.

**Author contributions**

FMSAK, HJB, SB and FP conceptualised this research. FMSAK, MOU and FP planned the fieldwork and carried it out with help from CRTM and SB. FMSAK, PCH, CRTM and MTJM performed laboratory analyses. FMSAK prepared the manuscript, HJB and FP revised and edited the draft with contributions from the other co-authors.

**Competing interests**

The authors declare that they have no conflict of interest.

**Acknowledgements**

This research was supported by Veni Grant 863.13.016 from the Dutch Research Council (NWO) to F.P. at Utrecht University (UU). We thank prof. Prasanta Sanyal for his guidance and help in planning the fieldwork in the Godavari basin. We thank dr. Maarten Lupker (ETH Zurich) and Huub Zwart (UU) for their assistance in the field. We are grateful to C. Mulder, D Kasjaniuk, A. van Leeuwen-Tolboom, D. van den Meent-Olieman and K. Nierop for technical and analytical support. We thank J.F. Veldkamp (Naturalis Biodiversity

Centre/National Herbarium of the Netherlands, Leiden) as expert on Southeast Asian botany for his help with taxonomical classification of the Indian peninsular plants. We are grateful to Prof. Sarah Feakins and an anonymous reviewer for their helpful comments, which greatly improved our manuscript.

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
