# Peer review of "Fig. S1: Zoom for Godavari delta, with sampling sites and the major vegetation zones (Olson et al., 2001; Asouti and Fuller, 2008)."

_Biogeosciences, 2022_

## Author Comment (AC1)

**Author's response to Reviewer#1 (anonymous)**

*General comments:*

The authors have worked with a variety of samples viz. vegetation, soil, suspended particulate matter, riverbed sediments from the Godavari region which is commendable.

**Response**: We thank the review for their positive assessment of the approach we used in this paper.

But I am curious as to why the authors chose isotopic analyses for the study. The authors must note that there are other much stronger techniques that can be applied for palaeovegetation reconstruction, such as geochemical biomarkers or compound specific isotopic analyses of normal alkanes. These provide much more detailed/spot on information without significantly less biases or overlaps. So the authors must signify and explain very clearly the selling point of this paper and strength of the technique that has been used.

**Response**: We agree with the reviewer that there are different techniques available for vegetation reconstructions, including plant-specific markers and/or compound-specific isotopic analysis, which are often applied in settings of marine sediments where a core is analysed that can give information about the past of the river basin. For a high-resolution study like we did in the Godavari basin with analysis of many modern plants, soils, riverbed and suspended sediments, bulk carbon isotope measurement provide a low-cost and high-throughput approach that can be easily applied in large geographic areas to determine the contribution of C3 and C4 vegetation. This is particularly important in arid tropical and subtropical basins, where carbon and carbon-based biomarker concentrations in soils and sediments are generally low and highly susceptible to degradation due to the high prevailing temperatures. Furthermore, sewage and waste inputs into (sub-)tropical rivers may contaminate a biomarker signal. For instance, compound-specific n-alkane analysis in the Godavari basin was not feasible due to low concentrations in soils and sediments and minor fuel spillages altering the n-alkane signal.
→ We will include this motivation for bulk carbon isotopic analysis in the introduction.

I would also like to add that unless a journal's terms and conditions require so, it is generally not a good idea to combine "results and discussion". Separating the two makes it much clearer as to what your own data and results are depicting and the discussion would include clear explanations of your results. It is important for readers to identify the original data of your research and separate them from previous literature data and knowledge that come under discussion part.

**Response**: We are aware that there are different ways to organise results and/or discussion sections and Biogeosciences does not require a certain format. Given our extensive and diverse dataset (vegetation, soils, SPM, and riverbed sediments for both the wet and the dry season), we used a combined results and discussion section to keep the overview of which part of the dataset is being interpreted. We believe that our format helps to create a good flow in the manuscript. For example, we need the results and discussion in each subsection (3.1: $\delta^{13}$C in C3 and C4 plants and controls by MAP) to move into the next section (3.2: $\delta^{13}$C in soils and sediments and their controlling factors) to build up to the application of a mixing model for C3/C4 contributions (3.3).
→ We will check the manuscript and emphasise our results by including phrases like: our results/ Godavari plants or samples show.

*Specific comments:*

Page 6, Line 150-152: There is no mention in the introduction as to how microbial inputs or early diagenetic alterations and early decomposition of organic matter might affect the isotopic signatures. The present approach is highly one-dimensional primarily considering input of C3 vs. C4 vegetation in connection with wetter and drier conditions. Authors must take into account all the other factors, particularly those which significantly influence isotopic fractionation.

**Response**: We agree that a complex interplay of different inputs and processes affect the organic matter (and its $^{13}C$ composition) in soils, including preferential degradation, carbon-containing contaminations, and differential fractionation by bacterial and fungal biomass. While the majority of soil organic matter derives from microbial processed plant residues, the actual size of the microbial biomass is difficult to ascertain and highly dependable on the availability of labile carbon sources and moisture levels. Other researchers estimated that (for conditions with sufficient moisture levels and easily available C) the microbial biomass itself contributes only 1-5% of the total soil organic matter (Kögel-Knabner, 2002; Simpson et al., 2007). Furthermore, there will be considerable spatial heterogeneity in microbial biomass distribution in field studies and techniques to accurately determine the microbial biomass size are challenging (Birge et al., 2015; Wiesmeier et al., 2019). Therefore, we consider determining the contribution of the microbial biomass in the Godavari basin beyond the scope of our study. We note that we sampled the Godavari soils at the end of the dry season, when moisture levels were very low and likely limited the size and activity of the microbial biomass.
→ We will include a note in the introduction on the different processes/inputs affecting soil organic carbon and its isotopic composition in soils, which becomes more positive due to microbial processing. In the discussion (3.2.1) on degradation processes in soils, we will expand on the role of the microbial biomass and include the issues raised above.

Page 8, Line 190: "aboveground plant parts" - Is any part of a plant other than leaves being sampled?

**Response**: Yes, for trees and large shrubs the leaves were sampled, and for grasses and herbs the leaves and stems were combined.
→ We will specify this in the MS.

Page 11: "Modern C3 and C4 plants in the Godavari basin and control by MAP" - Are there no CAM plants in the region?

**Response**: A literature search revealed that CAM plants are not wide-spread in India (Sankhla et al., 1975; Ziegler et al., 1981), and this corresponds to our field observations that succulents are not prevalent. Modern Indian plants on the Gangetic plains (Basu et al., 2015) and our study in the Godavari basin were all identified as species with C3 and C4 photosynthetic pathways.
→ We will include a note on the (limited) occurrence of CAM plants in India in the introduction.

Page 13, Line 306-308: "For C4 plants, the plants collected in the Godavari basin had significantly more negative δ13C values than the global average estimate (-14.0±0.2 ‰ (±standard error: SE) vs. -12.0 ‰; p≤0.001), revealing a difference between the local and global average C4 end-members" - Can you explain this observation?

**Response**: We analysed 16 C4 plants in the Godavari basin which fell within a relatively narrow isotopic range, and were all sampled at relatively dry locations with MAP mainly <1000 mm/y. Basu et al. (2015) also found relatively low $δ^{13}C$ values in C4 plants from the

Gangetic plains at < 1000 mm MAP, which they attributed to a moisture control resulting in lower $\delta^{13}C$ values in moisture-stressed C4 plants. However, most other studies found no correlation between C4 plant $\delta^{13}C$ and MAP. Potentially, the sample size of 16 C4 plants in the Godavari basin over a limited MAP range was too limited to identify a moisture control. In addition, correction of the Godavari vegetation dataset for the Suess effect, as suggested by both reviewers, may reduce the offset with global C4 plant data.

→ In the revised MS we will expand our discussion about a potential moisture control on C4 carbon isotopic values and discuss potential implications of the sample size of Godavari C4 plants. We will also consider the impact of the Suess correction for the comparison of C4 plant data.

Page 14, Line 317-318: Was the d13C corrected for Suess effect?

**Response**: The d13C data have not been corrected for the Suess effect. We thank the reviewer (as well as Prof. Feakins) for notifying this. We will correct d13C data in our manuscript where applicable. In this specific instance, the effect of MAP on C4 plant $\delta^{13}C$ values was evaluated and discussed for contemporary plants in the Godavari basin. The other studies cited here that are used for comparison also investigated modern plants within a certain precipitation range. We, therefore, believe that the comparison we make here is reliable and that the conclusions will hold after Suess correction. Regardless, we will check throughout the MS where contemporary measurements are compared with data reported in older literature, and correct for the Suess effect when applicable.

A preliminary assessment of the effect of the Suess correction on the comparison between our measurements on Godavari plants and data reported in literature, as well as for the plant end-members used in the mixing model shows that the overestimation of %C4 due to moisture availability will become slightly smaller (6% vs 10%). Moreover, there is still a substantial underestimation of the %C4 plants when using global plant end-members compared to those based on Godavari vegetation (15 vs 19%). Hence, the effect of Suess correction on the outcomes is relatively small, and our main conclusions about the mixing model appear to be still valid. We will take care to include Suess-corrected data and our full re-analysis thereof in the revised MS.

Page 15, Line 334-335: "This difference suggests that the latter value, which is reportedly strongly biased towards dry ecosystems" - This is not clear.

**Response**: This line will be rephrased to clarify.

Page 15, Line 336-337: "was significantly less negative in the upper basin (-28.0±0.3 ‰, n=32) than in the lower basin (-28.8±0.2 ‰, n=45; p≤0.05)" - Does this difference in the value qualify as "significantly" less?

**Response**: The difference between the upper and lower basin is indeed small, but significant.
→ We will rephrase these lines accordingly.

Page 15, Line 337: "reflecting the gradient in MAP" - Is the difference enough to conclude a "gradient in MAP"?

**Response**: Yes, MAP ranges from ~400 to 2300 mm/y based on long-term data. The difference in MAP between the upper and lower basin was previously reported in section 2.4.
→ We will relocate the information MAP in the upper and lower basin to this section to clarify the gradient in MAP.

Page 15, Line 342: "Pearson's R = -0.34" - This is not even significant especially for such a small population.

**Response**: The effect of MAP on $\delta^{13}C$ values of the individual C3 plants is indeed weak, but significant following an independent samples t-test. The significance is indicated by the two **.
→ We will specify this in the revised MS and include an explanation of the level of significance in the caption of Fig. 4.

Page 15, Line 350-352: "Moreover, the Godavari C3 plants were not evenly distributed over the entire precipitation range. Together, this resulted in a relatively weak linear correlation with MAP for the individually measured C3 plants" - This contradicts the previous sentence on line 341. If the effect of MAP on isotopic values are significant, shouldn't the correlation be high?

**Response**: The Pearson's R provides information about the strength and direction of the correlation between MAP and $\delta^{13}C$ values of the individual C3 plants, which is -0.34, due to the large variability we observe for the individual C3 plants at a certain level of MAP. Next, the linear correlation estimates the parameters that can be used to predict $\delta^{13}C$ values based on MAP. The $R^2$ represents the amount of variation in $\delta^{13}C$ values that can be explained by MAP (which is ~12 % for the C3 plants). A significant correlation between $\delta^{13}C$ values and MAP (as expressed by the significant Pearson's R) does not imply a high $R^2$ (i.e. the factor that explains most of the variation) in the linear regression.
→ We will clarify in the revised MS that the $R^2$ describes the amount of variation that can be explained by the independent variable (here MAP).

Page 16, Line 384-386: "This isotopic contrast corresponds with the vegetation distribution in the basin, with mixed C3 and C4 vegetation in the upper basin and more C3 plants in the lower basin" - Is the vegetational input only controlling factor for the isotopic values? What about any signatures of soil bacteria?

**Response**: Our observation of different soil $\delta^{13}C$ values the upper and lower basin points toward a control by the overlying C3/C4 vegetation. As discussed in the response to the reviewer's comment on Page 6, Line 150-152, the contribution of the microbial biomass is likely small (i.e., 1-5%). We agree that part of the soil organic carbon will be microbially processed, and potential effects of this degradation process are discussed in section 3.2.1
→ We will expand (see response Page 6, Line 150-152), the discussion on microbial processing effects on soil $\delta^{13}C$ values.

Page 18, Line 408-410: "However, C4-derived OC has also been shown to be preferentially incorporated into fine fractions where it is better protected against degradation, whereas C3-derived OC is preferentially added to the coarse fraction thus leaving it less protected" - What governs this affinity for the C4 plants towards finer fractions whereas C3 plants towards coarser fractions?

**Response**: The C4-derived OC has been shown to contain more labile, easy degradable compounds. The fine soil particles have presumably a higher ability to stabilise these labile, C4-derived compounds onto mineral surfaces than C3-derived OC that contains more difficult degradable compounds (partially derived from woody plant parts) that preferentially end up in the coarser soil fraction (Wynn, 2007; Wynn and Bird, 2007). In addition, the upper basin, where C4 contributions are highest, is characterized by soils that contain more clays than those in the lower basin, facilitating the enhanced association of C4 vegetation with mineral surfaces.

→ We will include information on the labile vs difficult degradable compounds contained in respectively C4- and C3-derived OC, resulting in a different stabilisation process in different soil fractions.

References

Basu, S., Agrawal, S., Sanyal, P., Mahato, P., Kumar, S. and Sarkar, A.: Carbon isotopic ratios of modern C3–C4 plants from the Gangetic Plain, India and its implications to paleovegetational reconstruction, Palaeogeogr. , Palaeoclimatol. , Palaeoecol., 440, 22-32, 2015.

Birge, H. E., Conant, R. T., Follett, R. F., Haddix, M. L., Morris, S. J., Snapp, S. S., Wallenstein, M. D. and Paul, E. A.: Soil respiration is not limited by reductions in microbial biomass during long-term soil incubations, Soil Biol. Biochem., 81, 304-310, 2015.

Kögel-Knabner, I.: The macromolecular organic composition of plant and microbial residues as inputs to soil organic matter, Soil Biol. Biochem., 34, 139-162, 2002.

Sankhla, N., Ziegler, H., Vyas, O. P., Stichler, W. and Trimborn, P.: Eco-physiological studies on Indian arid zone plants, Oecologia, 21, 123-129, 1975.

Simpson, A. J., Simpson, M. J., Smith, E. and Kelleher, B. P.: Microbially derived inputs to soil organic matter: are current estimates too low?, Environ. Sci. Technol., 41, 8070-8076, 2007.

Wiesmeier, M., Urbanski, L., Hobley, E., Lang, B., von Lützow, M., Marin-Spiotta, E., van Wesemael, B., Rabot, E., Ließ, M. and Garcia-Franco, N.: Soil organic carbon storage as a key function of soils-A review of drivers and indicators at various scales, Geoderma, 333, 149-162, 2019.

Wynn, J. G.: Carbon isotope fractionation during decomposition of organic matter in soils and paleosols: implications for paleoecological interpretations of paleosols, Palaeogeogr. , Palaeoclimatol. , Palaeoecol., 251, 437-448, 2007.

Wynn, J. G. and Bird, M. I.: C4-derived soil organic carbon decomposes faster than its C3 counterpart in mixed C3/C4 soils, Global Change Biol., 13, 2206-2217, 2007.

Ziegler, H., Batanouny, K. H., Sankhla, N., Vyas, O. P. and Stichler, W.: The photosynthetic pathway types of some desert plants from India, Saudi Arabia, Egypt, and Iraq, Oecologia, 48, 93-99, 1981.

---

## Author Comment (AC2)

**Author's response to Reviewer#2 (Sarah Feakins)**

The study is interesting, and would likely be suitable for Biogeosciences after moderate revision.

**Response**: We thank the Prof. Feakins for her positive overall assessment of the paper.

*Summary points:*

I agree with the other reviewer that I was surprised to find this was a bulk OC carbon isotope based study. In fact, I assumed the study was based on plant wax when I accepted the request to review based upon the paper coming from a biomarker lab. Of course using bulk methods doesn't invalidate the study, but it could be more clearly signaled for the reader. Using the word "bulk" at first reference to on line 32 "In this study we investigated the bulk carbon isotopic signature.." may suffice.

**Response**: This is a good suggestion, we will add the word 'bulk' in the abstract. We will also carefully consider the use of 'bulk carbon isotopic measurements' throughout the revised MS.

The authors appear to have neglected the changing atmospheric d13C over recent decades and how that would affect carbon in modern plants, and potentially older soils and fluvial SPM. Literature comparisons span 1970s to present and it needs accounting for. Please add discussion of (likely) age of materials and the Suess effect, throughout wherever relevant, and account for this numerically.

**Response**: We would like to thank Prof Feakins for pointing this out. We are actually amazed by the fact that none of the (co-)authors has thought about this issue. We will correct our data for the Suess effect, redo the analyses, and adjust the manuscript where appropriate. Note that we do not suspect our conclusions on vegetation to change, as all vegetation was collected during the same expedition and plant $\delta^{13}C$ values will thus be corrected to the same extent, leaving the trends with MAP reported in the manuscript intact. The same would be true for the correction of bulk soil $\delta^{13}C$, as soils in the upper and lower part of the Godavari basin are of similar age (Usman et al., 2018). We will also explore the implications of the Suess correction for the mixing model outcomes.

We did a preliminary assessment of the effect of the Suess correction on the comparison between our measurements on Godavari plants and data reported in literature, as well as for the plant end-members used in the mixing model. The Suess correction does not seem to affect our main conclusions. The mixing model outcomes are still valid, with a substantial underestimation of the %C4 plants when using global plant end-members compared to those based on Godavari vegetation. Without question, we will take care to include Suess-corrected data in the revised MS.

There is some duplication of graphs Figs 1c and 2a, Figs 2b and 3a – that would ideally be organized so that data are only presented only once graphically.

**Response**: We agree with the reviewer that the cited figures are superfluous. We will reconsider them and combine information in these figures as requested.

*Detailed line by line comments follow:*

Line 45 "Our analysis revealed that the reconstructed C3/C4 vegetation composition was sensitive to the plant δ13C end-members used as mixing model input." Please do not frame this as a new 'analysis revealed' as it is well known that the C3 'endmember' is a flawed

concept as it has a very wide spread. Informed choices about more meaningful endmembers may be possible in some instances e.g. if it is known to be wet rainforest or dry C3 desert for example. However there have been other attempts to work around this mathematically including the Fwoody cover approach with a nonlinear fit [1].

**Response**: We agree with the reviewer that mixing models provide not perfect representation of the spatial C3/C4 distribution in a region and that it is important to make better informed choices about meaningful end-members. To the latter end, we explored how the use of different plant end-members and (correction for) the amount of MAP in a region influences the outcomes %C4 and C3 plants by a mixing model. We used a validated linear isotope mixing model for this (see Philips and Gregg, 2001), that includes source variability and error propagation, thus minimising the spread in mixing model outcomes. Such linear mixing models have been and are used worldwide for C3/C4 vegetation reconstructions, and we address a range of studies in the Introduction. We are aware that there are alternative methods such as the non-linear Fwoody approach to estimate C3/C4 distributions, that focuses on the vegetation structure and canopy shading effects. We argue, however, that for the Godavari basin, and many basins with intense human influence, such an approach is complicated by the fact that the natural vegetation has been replaced by agriculture resulting in a more open landscape (e.g. Wynn and Bird, 2008). For the Godavari basin, intensification of agriculture over the Holocene and deforestation since the 19[th] century mean that over 60% of the basin is used for agriculture, and only a small area is still covered by native (tropical) forest which would be sensitive to changes in Fwoody C3 vegetation.
--> We will rephrase Line 45 in the abstract to focus on the results of the C3/C4 reconstruction in the Godavari basin and not frame this as a new concept. In addition, we will incorporate the potential influence of the canopy effect on our vegetation d13C data in the discussion section.

Godavari specific endmembers, this would be more generally interesting if we were told right away if this is the wet or dry end of C3 etc, unlikely that there are regional plant species effects, likely it is just the usual canopy etc effects.

**Response**: We agree that extra information on where the Godavari basin can be placed in terms of dry or wet conditions is helpful to put our findings into a global perspective. Note, however, that we do not consider the canopy effect to play a significant role in the Godavari Basin (see reply to previous comment).
--> We will add information in the abstract about the MAP range in the Godavari basin. In addition, in the Methods (section 2.4) we will add information on Godavari basin and global occurring C3 vegetation in relation to MAP.

Line 49 " Hence, incorporating region-specific plant δ13C end-members and drought correction of the C3 end-member in mixing models need to be considered to determine C3 and C4 distributions of modern- and paleo-vegetation in monsoonal regions." Rephrase this sentence.

**Response**: We will rephrase these lines to reflect the discussion on the use of different plant end-members for vegetation reconstructions.

Line 56 – all the cited references refer to bulk plant tissue and are references that span 1970-2010. The difference in plant d13C between 1970 and 2020 is ~2 per mil. Please check and see what a recent collation of data has reported after correction for the date of collection, or do the work to update this to a consistent modern value suitable for comparison to your plants. Your soils and river samples may integrate more time however and thus the temporal shift may also be relevant to summarize here in the introduction.

**Response**: We will correct the $\delta^{13}C$ bulk plant tissue data from the cited papers for the Suess effect to allow comparison with the data we measured in the Godavari basin. We will also consider the age of the soils and riverbed sediments ($^{14}C$ data from Usman et al., 2018 for a selection of samples and soil turnover times reported in the literature) and correct the plant end-members accordingly for the mixing model. This information will be included in the methods section as well as the discussion.

Line 89 – the concept of endmembers is flawed, especially for C3, instead it is important to describe the spread of C3 plants as context for any central estimate. This section of text is also flawed in that it misses the timescale of sampling. Internal to a study the C4 response to dryness has been found to be quite small 1 per mil (Cerling) not absent as concluded in this plant study in the results section, but perhaps the n is too small to be sure?

**Response**: We will provide additional argumentation on our choice to use an endmember model for our study, also including the aspects of fraction woody cover, and 'wet' and 'dry' C3 endmembers, and the timescale of sampling, as explained in our response to the earlier comment on Line 45. Note that we deliberately selected a mixing model that includes source variation and error propagation to overcome most of the above issues. As responded to Reviewer#1, we will expand on the impact of MAP on C4 plants and consider the sample size of the C4 plants we evaluated in the Godavari basin for a control by MAP.

Line 94 is on C4, then line 95 returns to C3 again, and another switch is found later on – the flow needs organizing.

**Response**: We will reorganise this paragraph to establish a good flow, and focus first on C3 and then on C4 plants.

Line 110 – I do not find the concept of a 'global average' C3 plant d13C to be useful.

**Response**: We will reconsider this concept after application of the Suess correction and see what the implications are for the difference in $\delta^{13}C$ data we measured in the Godavari basin and the commonly quoted 'global averages' for C3 and C4 vegetation.

Line 111 – regional average is also not very useful, more useful to think in terms of the vegetation category average e.g. closed forest, open woodland etc.

**Response**: We introduced the term Godavari-based or regional end-members to refer to (weighted) $\delta^{13}C$ values of C3 and C4 vegetation that evolve from the specific habitat conditions in the Godavari basin, which likely results from a combined effect of the (Indian) plant species, vegetation structure (density, distribution, canopy etc.), moisture, temperature and other environmental conditions. The field survey we did in the Godavari basin does not provide controlled conditions where we could assess the individual impact of these factors on vegetation $\delta^{13}C$ values. Hence, we use the term regional or Godavari-based.
→ We will include this motivation in the revised MS in the introduction, and in the Methods (section 2.5) we will define regional and global. Notably, we will re-evaluate the comparison between Godavari and global end-members after the Suess correction, and revise this section if appropriate.

Line 116-119 – sentence needs revisiting – rephrase. Note that this refers to a study that is also conceptually based on the endmember approach.

**Response**: We will rephrase these lines and include information that this result was based on an end-member approach which used specific plant $\delta^{13}C$ values based on samples of the regional vegetation.

Consider moving away from the outdated concept of a C3 endmember and moving to something like the non linear Fwoody cover approach that deals with the issues of spread in C3 plants. Or if you insist upon a linear mixing model make sure you propagate the uncertainties caused by the C3 distribution upon those C4% estimates. If you do error propagation, you'll see the issue.

**Response**: As explained in our response to your earlier comment on Line 45, we used the isotope mixing model from Philips and Gregg (2001) that includes the variability in the C3 and C4 vegetation (i.e., sources) as well as in the soils (i.e., mixture), with consideration of the uncertainties and error propagation in isotope mixing. As discussed before, the Fwoody approach is complicated by agriculture in the Godavari basin, which has created a more open landscape and drastically limited the area with native forest vegetation. As a result, we believe that the variation in the Godavari vegetation d13C cannot be attributed to changes in Fwoody cover, but are driven by changes in moisture availability instead.

Paragraph beginning 121 discusses plant to soil to river degradation fractionations well. It neglects to discuss the age of the OC and the Suess effect means that 2 per mil needs to be accounted for when comparing today's plants and a couple decade old OC in soil/sediment. Old OC would be 2 per mil more enriched compared to today's OC without any degradation fractionation.

**Response**: Thanks again for pointing this out. We will correct all our data for the Suess effect to enable a fair comparison between vegetation, soil, and sedimentary data, and incorporate subsequent changes in the revised version.

Line 153 why (paleo-)vegetation reconstructions? "vegetation reconstructions" suffices. Same issue throughout e.g. line 586 and conclusion title line 589.

**Response**: We will change this in the revised manuscript.

*Methods*

Plant and river sampling methods are appropriate and well described. The only question I'm left with is are the plant samples representative, when sampling bulk from a tree, the trunk is the bulk of the biomass, although the production of leaves may have a faster rate. When sampling leaf wax the leaves are appropriate, but when sampling bulk is the leaf sampling appropriate? I can see it is hard to homogenize a tree unlike sampling grasses (or leaf waxes) where the sampling task is simpler.

**Response**: See also our response to Reviewer#1 on this issue. Since all trees are deciduous, and shed their leaves annually, we have assumed that the leaves are the main contributor of OC to the soils, rather than the trunk. As for the representation of the plant species, we have collected the ~5 most common species at each site. We also made sure to select those species that were present in large(r) parts of the basin to be able to study the effect of MAP on their stable carbon isotopic composition.

Line 236 "robust relationship between MAP and d13C has been shown to prevail in C3 plants around the world" yes there is a trend but also a lot of scatter. This is acknowledged on line 249 a long way after for the reader, and the solution we are told is "binning'" on line 249 but binning is not explained, that I have found in the text.

**Response**: We will move the scatter in the relation between d13C and MAP up in the manuscript, and also better explain the approach that was used to bin our data.

*Results and Discussion*

Line 289 the plants falling in the "lower" end of the global range is consistent with the comparison of modern plants and an older global literature reference comparison. However just on numerical comparison "lower end" also seems to be a misrepresentation as closed tropical forest would be lower. Reconsider.

**Response**: We will reconsider this part of the discussion after we have corrected our data for the Suess effect. As mentioned earlier, the fraction woody cover is unlikely to play an important role in the Godavari basin, but we will add the explanation of why we believe that this is so to the revised manuscript.

Line 354 – binning – apologies, if I've missed it but I don't see this explained yet, and so I struggle to follow this.

**Response**: We will clarify our binning approach in the revised version of the manuscript. This information was briefly summarised in the caption of Fig 4, but will move this to method section 2.4 where the regression analysis is explained, and expand this information with more details about our approach.
--> We will include the following information: In order to deal with the uneven distribution of Godavari C3 plant $\delta^{13}C$ values over the MAP range, we applied a binning approach. The data were binned by calculating the average and standard error of C3 plant $\delta^{13}C$ values per MAP range of 100 mm $y^{-1}$. These binned Godavari C3 plant data were subsequently plotted against the average MAP of each respective range and utilised for regression analysis to assess the relation between C3 plant $\delta^{13}C$ values and MAP.

Line 439 remove "interestingly" which is subjective, and this well-known issue is one reason why reviewer 1 questioned the use of bulk, it becomes problematic in estuarine and marine settings as is well known (and perhaps no longer that interesting).

**Response**: We will make this change and rephrase this sentence where appropriate.

Line 529 – though we found some wood far offshore in the Bengal Fan [2]

**Response**: The Bengal Fan is fed by Ganges-Brahmaputra system can indeed carry coarse, woody materials far offshore (Lee et al., 2019), due to the high sediment transport potential with high flow velocities in the wet season along a relative steep altitudinal gradient from the Himalayas to the Bay of Bengal. In contrast, rivers with a limited altitudinal gradient such as the peninsular Indian rivers and the Godavari system appear to be more susceptible to depth-related sorting during river transport. As such, no wood was found in Holocene cores sampled in front of the Godavari mouth (Ponton et al., 2012; Giosan et al., 2017; Usman et al., 2018). Of course, this absence does not exclude the transport of wood by the Godavari River, but based on the shape of the river basin we believe that this transport will be limited compared to that by large(r) rivers, such as the Ganges-Brahmaputra.
--> We will include this information in the MS and specify the difference between the Godavari River system and the Bengal Fan that is fed by the Himalayan-derived Ganges-Brahmaputra.

*Conclusions*

Line592 – the discussion makes it sound like there is something regionally unique about the d13C when they fall within the global plants dataset and likely overlap with similar vegetation types. Thus it is more vegetation type/habitat/MAP considerations rather than geographic regions that should be emphasized, and so doing would make it more globally of interest than local.

**Response**: We will re-assess the differences between our Godavari dataset and the global compilation after we have corrected all data for the Suess effect. We expect that the Godavari dataset will be more similar to the global data after the Suess correction. Consequently, we will revise this section on the local vs global comparison if needed.

*Figures:*

Fig 1 – the map figures are useful, for the third panel showing MAP is the partition of the upper and lower basin based on the MAP, if so or otherwise, please give the numerical basis for the partition in the caption for this panel. Preferably change to a green-brown or blue saturations color scale rather than rainbow to be intuitive visually, and provide a legend that can be read in a quantitative sense, see comment on Fig 2a). Please note the repetition of data visuals, Fig 1c and Fig 2a are duplicative. Duplication should be removed. Fig 1c can be removed, as 2a conveys data at the site sampling points as well as the basemap.

**Response**: We will include information on the partition between the upper and lower basin, which is based on a difference in geology which also influences the vegetation, moisture availability and land use types – we will emphasise this difference in the method section 2.1 describing the regional setting. We will investigate if we can change the colour scheme to a green/brown/blue scale or a saturation of a single colour. Since Fig.1c and 2a give the same information, we will remove Fig 1.c, as requested.

Fig 2 a) apart from other concerns regarding the rainbow color scheme that have been widely reported, I would also not encourage the use of scale bar that is purely qualitative for the MAP data. It is not possible to read between the numbers 430 and 2300 mm/yr and know what 'yellow' or 'green' represents in terms of MAP. You can use a scale with incremental output and a color scheme that is a saturation of a single color which will help to allow for visual quantitative evaluation of where is wetter and drier.

**Response**: As responded for Fig 1, we will look into a different colour scheme to display MAP, preferably a saturation of a single colour, to improve the readability of the scale and better visualise drier and wetter areas.

d13C data points with the rainbow colors can be discerned by most readers using the legend, the coloring is not intuitive, for wet to dry try green to brown for example, and it would be better to pick a color scheme that can be seen by all readers.

**Response**: We will also look into a different colour scheme for the points depicting $\delta^{13}$C values in Fig 2.

b) Why are upper and lower basins parsed. Are these much different, probably not as the C4 distribution in lower basin falls within that for the upper basin, and the same for C3 with the upper basin just having a bit more range. Maybe overlay the two bar charts or use violins, to display the data if you want to keep with this 2 category, but if you do an T or F test do you find they are significantly different? (this panel is repeated in fig 3) fig. 2b can therefore be deleted.

**Response**: As responded to the earlier comment on Fig 1, the upper and lower basin of the Godavari are distinctly different in terms of underlying geology, C3/C4 vegetation distribution, MAP, and land use types. We will expand on this in method section 2.1 describing the regional setting. The upper and lower basin are compared throughout the MS, significant differences in $\delta^{13}$C values are revealed in Fig 3 and discussed thereafter.

Fig 3 – shows a bar chart of the same data as in figure 2b but in box and whisker format. The data only need to be shown once. As this plot is better this is the plot that should be retained and 2b deleted.

**Response**: We will remove the histogram in Fig 2b, as requested, and only keep Fig 3 which depicts the data most clearly.

Fig 4 – why show 'global C4' as a line = -12 per mil. Where does this derive from? Is it the mean of a collection of plants over several decades, without representation of the scatter in that dataset or correction for the accelerating d13C change in atmospheric CO2 over the last 2 decades. I assume your plants are simply showing scatter consistent with the global dataset, after correction for atmospheric d13C and pCO2 change over time.

**Response**: The global C4 and C3 lines represent commonly quoted 'average' $\delta^{13}C$ values reported in literature studies. We will investigate if a Suess correction is possible to update these values to modern conditions and look into the implications. Since the scatter in these data is not reported, we used the observed variation in the Godavari C3 and C4 plants when using these global averages as input for the mixing model.

References

Giosan, L., Ponton, C., Usman, M., Glusztajn, J., Fuller, D. Q., Galy, V., Haghipour, N., Johnson, J. E., McIntyre, C. and Wacker, L.: Massive erosion in monsoonal central India linked to late Holocene land cover degradation, Earth Surf. Dynam., 5, 781–789, 2017.

Lee, H., Galy, V., Feng, X., Ponton, C., Galy, A., France-Lanord, C. and Feakins, S. J.: Sustained wood burial in the Bengal Fan over the last 19 My, PNAS, 116, 22518-22525, 2019.

Phillips, D. L. and Gregg, J. W.: Uncertainty in source partitioning using stable isotopes, Oecologia, 127, 171-179, 2001.

Ponton, C., Giosan, L., Eglinton, T. I., Fuller, D. Q., Johnson, J. E., Kumar, P. and Collett, T. S.: Holocene aridification of India, Geophys. Res. Lett., 39, L03704-L03709, 2012.

Usman, M. O., Kirkels, F. M. S. A, Zwart, H. M., Basu, S., Ponton, C., Blattmann, T. M., Ploetze, M., Haghipour, N., McIntyre, C. and Peterse, F.: Reconciling drainage and receiving basin signatures of the Godavari River system, Biogeosciences, 15, 3357-3375, 2018.

Wynn, J. G. and Bird, M. I.: Environmental controls on the stable carbon isotopic composition of soil organic carbon: implications for modelling the distribution of C3 and C4 plants, Australia, Tellus B Chem. Phys. Meteorol., 60, 604-621, 2008.

---

## Author Response (AR1)

Utrecht, 12 July 2022

Dear editor,

We have revised our manuscript "Carbon isotopic ratios of modern C3 and C4 vegetation on the Indian Peninsula and changes along the plant-soil-river continuum; Implications for vegetation reconstructions" based on the comments of Prof. Sarah Feakins and an anonymous referee.

We thank the reviewers for their feedback and have followed most of their suggestions, as you can read in our replies posted on the forum and our list of point-to-point changes below. All changes in the manuscript are marked in red in the track change version.

We hope that you find this revised version suitable for publication in Biogeosciences.

On behalf of all co-authors,

Frédérique Kirkels

**Point-by-point reply for Reviewer#1 (anonymous)**

General comments:

The authors have worked with a variety of samples viz. vegetation, soil, suspended particulate matter, riverbed sediments from the Godavari region which is commendable.

*We thank the reviewer for their positive assessment of the approach we used in this paper.*

But I am curious as to why the authors chose isotopic analyses for the study. The authors must note that there are other much stronger techniques that can be applied for palaeovegetation reconstruction, such as geochemical biomarkers or compound specific isotopic analyses of normal alkanes. These provide much more detailed/spot on information without significantly less biases or overlaps. So the authors must signify and explain very clearly the selling point of this paper and strength of the technique that has been used.

*Reply: We agree with the reviewer that there are different techniques available for vegetation reconstructions, including plant-specific markers and/or compound-specific isotopic analysis, which are often applied in settings of marine sediments where a core is analysed that can give information about the past of the river basin. For a high-resolution study like we did in the Godavari basin with analysis of many modern plants, soils, riverbed and suspended sediments, bulk carbon isotope measurement provide a low-cost and high-throughput approach that can be easily applied in large geographic areas to determine the contribution of C3 and C4 vegetation. This is particularly important in arid tropical and subtropical basins, where carbon and carbon-based biomarker concentrations in soils and sediments are generally low and highly susceptible to degradation due to the high prevailing temperatures. We included this information in the introduction to motivate our approach.*

*This now reads: "For vegetation reconstructions, different techniques can be employed, including bulk or compound-specific isotope analyses. Although less source specific, bulk isotope analyses provide a low-cost, high throughput approach that can be applied at high resolution and/or large geographic areas, also in (sub-)tropical regions where carbon and vegetation-specific compound concentrations are generally low and may undergo compound-specific degradation patterns and/or differential settling into sediments (e.g., Hou et al., 2020; Li et al., 2020). Subsequently, isotope mixing models provide a means to infer the distribution of C3 and C4 plants and changes therein at spatial or temporal scales, which is particularly relevant in context of changing climatic conditions affecting C3/C4 vegetation patterns. For instance, gradual aridification in a basin can result in drought-stressed C3 plants, as well as increased abundance of aridity-adapted C4 plants. Both changes result in an increase in bulk δ13C org values. A shift to more dominant C4 vegetation is important to identify as a shift to a dry ecosystem indicates a reduced resilience to changes in moisture availability (e.g., Cui et al., 2017; Ghosh et al., 2017)."*

I would also like to add that unless a journal's terms and conditions require so, it is generally not a good idea to combine "results and discussion". Separating the two makes it much clearer as to what your own data and results are depicting and the discussion would include clear explanations of your results. It is important for readers to identify the original data of your research and separate them from previous literature data and knowledge that come under discussion part.

*Reply: We are aware that there are different ways to organise results and/or discussion sections and Biogeosciences does not require a certain format. Given our extensive and diverse dataset (vegetation, soils, SPM, and riverbed sediments for both the wet and the dry season), we have used a combined results and discussion section to keep the overview of which part of the dataset is being discussed and interpreted. We believe that our format*

*helped to create a good flow in the manuscript. For example, we needed the results and discussion in each subsection (3.1: $\delta^{13}C$ in C3 and C4 plants and controls by MAP) to move into the next section (3.2: $\delta^{13}C$ in soils and sediments and their controlling factors) to build up to the application of a mixing model for C3/C4 contributions (3.3). Hence, we have maintained the structure of our manuscript as is.*

Specific comments:

Page 6, Line 150-152: There is no mention in the introduction as to how microbial inputs or early diagenetic alterations and early decomposition of organic matter might affect the isotopic signatures. The present approach is highly one-dimensional primarily considering input of C3 vs. C4 vegetation in connection with wetter and drier conditions. Authors must take into account all the other factors, particularly those which significantly influence isotopic fractionation.

*Reply: We agree that a complex interplay of different inputs and processes affect the organic matter (and its $^{13}C$ composition) in soils, including preferential degradation, carbon-containing contaminations, and differential fractionation by bacterial and fungal biomass. While the majority of soil organic matter derives from microbial processed plant residues, the actual size of the microbial biomass is difficult to ascertain and highly dependable on the availability of labile carbon sources and moisture levels. Other researchers estimated that – for conditions with sufficient moisture levels and easily available C – the microbial biomass itself contributes only 1-5% of the total soil organic matter (Kögel-Knabner, 2002; Simpson et al., 2007). Notably, we sampled the Godavari soils at the end of the dry season, when moisture levels were very low and likely limited the size and activity of the microbial biomass. Furthermore, there will be considerable spatial heterogeneity in microbial biomass distribution in field studies and techniques to accurately determine the microbial biomass size are challenging (Birge et al., 2015; Wiesmeier et al., 2019). Therefore, we considered determining the contribution of the microbial biomass in the Godavari basin beyond the scope of our study. In the introduction and discussion (3.2.1) we expanded on the role of microbial processing with respect to OC degradation and its potential impact on the carbon isotopic composition of our Godavari dataset.*

*This now reads in the introduction: "Furthermore, it is well-established that soil degradation processes enrich OC isotopes, which is usually estimated to be ~1– 3 ‰ but can be as high as 6 ‰ in tropical and semi-arid regions (e.g., Krull et al., 2005). Possible factors that contribute to this enrichment are preferred uptake and degradation to CO2 of 13C-depleted OC by microbes, incorporation of 13C-enriched microbial and fungal biomass in the soil and/or preferential adsorption of 13C by fine mineral particles (Krull et al., 2005; Wynn, 2007; Wynn and Bird, 2007). Soil OC thus comprises a mixture of plant-derived, fungal and bacterial biomass and microbially processed carbon. The different compounds (e.g., lipids, proteins, carbohydrates, etc.) differ in their degradability, but may also be associated with mineral surfaces, which protects them from degradation. Compound-specific degradation rates or preservation in soils via microbial processing or association with mineral particles may influence the reconstructed C3/C4 vegetation balance depending on the targeted compound, as shown for vegetation and soils in the Gangetic plain (Sarangi et al., 2021; Roy and Sanyal, 2022). This complex interplay between different inputs and microbial processing may challenge the use of stable carbon isotope ratios for vegetation reconstructions."*

Page 8, Line 190: "aboveground plant parts" - Is any part of a plant other than leaves being sampled?

*Reply: We added information on the exact plant parts that were sampled and analysed, as well as a motivation for our approach (following a comment from reviewer #2 Prof. Feakins)*

including the following line: *"For shrubs and trees the leaves were collected and for herbs and grasses the leaves and stems were combined".*

Page 11: "Modern C3 and C4 plants in the Godavari basin and control by MAP" - Are there no CAM plants in the region?

**Reply:** *We included information on CAM plants in the introduction, and referred to the fact that CAM plants have a limited occurrence of in India, based on studies by Sankhla et al. (1975) and Ziegler et al. (1981). Since all the plants included in our study were identified as C3 or C4 plants, CAM plants are not further considered in our manuscript.*

*This now reads: "Plants using an alternative photosynthetic pathway i.e., crassulacean acid metabolism (CAM) as adaption to aridity photosynthesise during the day and respire at night (i.e., temporal $CO_2$ concentrating mechanism), where moisture availability determines the expression of the C3 or C4 fixation pattern (Sankhla et al., 1975). Under water-stressed conditions CAM plants have isotopic values similar to C4 plants, but they are relatively rare in India (Sankhla et al., 1975; Ziegler et al., 1981) and are, therefore, not further considered here."*

Page 13, Line 306-308: "For C4 plants, the plants collected in the Godavari basin had significantly more negative $\delta^{13}C$ values than the global average estimate (-14.0±0.2 ‰ (±standard error: SE) vs. -12.0 ‰; p≤0.001), revealing a difference between the local and global average C4 end-members" - Can you explain this observation?

**Reply:** *We elaborated on the effect of moisture on $\delta^{13}C$ values in C4 plants in Results/Discussion section 3.1. We analysed 16 C4 plants in the Godavari basin which fell within a relatively narrow isotopic range, and were all sampled at relatively dry locations with MAP mainly <1000 mm/y. Basu et al. (2015) also found relatively low $\delta^{13}C$ values in C4 plants from the Gangetic plains at < 1000 mm MAP, which they attributed to a moisture control resulting in lower $\delta^{13}C$ values in moisture-stressed C4 plants. However, most other studies found no correlation between C4 plant $\delta^{13}C$ and MAP. We included a summary of these contrasting results in the revised MS. We further highlight that the sample size of 16 C4 plants in the Godavari basin over a limited MAP range may have been insufficient to observe a correlation with between C4 plant $\delta^{13}C$ values and MAP. Notably, we also mention here that correcting the C4 literature data for the Suess effect to match our Godavari data, reduced the offset between the Godavari C4 plants and global C4 vegetation, resulting in a small but still significant difference.*

*This now reads: "The Godavari C4 plants we sampled had on average more negative δ13C values than those collected in the only other extensive field survey of Indian plants on the Gangetic plain (-14.0±0.2 ‰ (±standard error: SE), n=16, (Fig. 3a) vs -12.7±0.2 ‰, n=45; p≤0.001), where they found most depleted signatures in areas with MAP <1000 mm y-1 and observed an effect of MAP on C4 plant δ13C values (Basu et al., 2015). In contrast, the Godavari C4 plants showed no significant correlation with MAP (Eq. 3; Pearson's R = -0.10; p=0.70) (Fig. 4),. The absence of a correlation for the Godavari C4 plants may be influenced by the relatively small sample size and their main occurrence in only a limited part of the MAP range covered (i.e., ~500 – 900 mm y-1 in 2014). Nevertheless,  earlier studies also predominantly found  no trends in C4 plant δ13C values in response to MAP in dry ecosystems around the globe (<800 mm y-1; Schulze et al., 1996; Swap et al., 2004). This finding was attributed to the $CO_2$ concentrating mechanism in C4 plants. This adaption to water loss due to evaporation in warm and dry climates may be influenced by leaking of $CO_2$ from bundle sheath cells during extreme drought, but functions relatively robustly for a wide range of environmental conditions, including drought stress (Murphy and Bowman, 2009). Taken together, we interpret that there was no basis for correction of  the δ13C C4 end-member for drought conditions."*

Page 14, Line 317-318: Was the d13C corrected for Suess effect?

*Reply: We now report Suess-corrected data in the revised manuscript, as well as the results of a full re-analysis thereof. In this specific instance, the effect of MAP on C4 plant $\delta^{13}C$ values was evaluated and discussed for contemporary plants in the Godavari basin. The other studies cited were only used to compare trends (not absolute $\delta^{13}C$ values). Hence, the comparison we made still holds, and also our main conclusions remain the same after Suess correction. For comparison of our Godavari plant data with those in (older) literature, we applied a correction for the Suess effect (see Table 1), as requested by both reviewers. We showed that after Suess correction of the plant end-members used in the mixing model approach, the overestimation of %C4 due to moisture availability was slightly smaller (6% vs 10%). However, there was still a substantial underestimation of the %C4 plants when using global plant end-members compared to those based on Godavari vegetation (15 vs 19%). Hence, the effect of Suess correction on our outcomes was relatively small, and does not affect our main conclusions about the mixing model setup and outcomes.*

Page 15, Line 334-335: "This difference suggests that the latter value, which is reportedly strongly biased towards dry ecosystems" - This is not clear.

*Reply: This sentence was superfluous and therefore removed in the revised MS.*

Page 15, Line 336-337: "was significantly less negative in the upper basin (-28.0±0.3 ‰, n=32) than in the lower basin (-28.8±0.2 ‰, n=45; p≤0.05)" - Does this difference in the value qualify as "significantly" less?

*Reply: We specified that the difference between the upper and lower basin is small, but significant.*

Page 15, Line 337: "reflecting the gradient in MAP" - Is the difference enough to conclude a "gradient in MAP"?

*Reply: We now explicitly mention the range in MAP in the abstract, and also made it more prominent in the site description (section 2.1). In addition, we relocated information on MAP that was previously in Methods section 2.4 to Results/Discussion section 3.1.*

*We included the following information to provide further context: "This finding corresponds to the observed spatial gradient in MAP in the Godavari basin and suggests an effect of MAP on C3 plant $\delta^{13}C$ values. Indeed, long-term MAP (1901–2015) was markedly lower in the upper than the lower Godavari basin (p≤0.001) and this contrast became more extreme for the 5-year average and 2014 MAP, which resulted in drought conditions in the upper basin (Fig. S2)."*

Page 15, Line 342: "Pearson's R = -0.34" - This is not even significant especially for such a small population.

*Reply: We reconsidered this and reported that the effect of MAP on $\delta^{13}C$ values of the individual C3 plants is indeed weak, but significant following an independent samples t-test. We included an overview of the different levels of significance marked with stars (*) in the caption of Fig. 4.*

*This now reads: "The individual Godavari C3 plants revealed a small, but significant effect by MAP on their δ13C values (Eq. 4; Pearson's R = -0.34; p≤0.0.1) (Fig. 4)."*

Page 15, Line 350-352: "Moreover, the Godavari C3 plants were not evenly distributed over the entire precipitation range. Together, this resulted in a relatively weak linear correlation with MAP for the individually measured C3 plants" - This contradicts the previous sentence on line 341. If the effect of MAP on isotopic values are significant, shouldn't the correlation be high?

***Reply:*** *The Pearson's R provided information about the strength and direction of the correlation between MAP and $\delta^{13}C$ values of the individual C3 plants, which was -0.34 and thus relatively weak, likely due to the large variability that we observe for the individual C3 plants at a certain level of MAP. The interpretation of this finding was explained in more detail section 3.1 of the revised MS. On the other hand, the linear correlation estimated a control by MAP on $\delta^{13}C$ values, where the latter could be predicted based on MAP. We included an explanation that the $R^2$ represented the amount of variation in $\delta^{13}C$ values that could explained by MAP. This was specified to be ~12% for the individual C3 plants, and ~82% for the binned C3 plants.*

*This now reads: "For the individual Godavari C3 plants, we noted considerable variation in δ13C values for any certain amount of precipitation, in line with earlier studies that found high inter- and intraspecies variation in C3 plant δ13C values in response to MAP (Ma et al., 2012; Liu et al., 2013, 2014; Basu et al., 2021; Luo et al., 2021). Moreover, the individual Godavari C3 plants were not evenly distributed over the entire precipitation range, making it more difficult to establish a correlation. Together, this resulted in a relatively weak linear correlation between MAP and individually measured C3 plants (Eq. 4; R2 = 0.12), where MAP explained only ~12% of the variation in C3 plant δ13C values. Subsequent binning of C3 plant δ13C values to overcome their uneven distribution over the range of MAP, revealed a strong and significant correlation with MAP (Eq. 7; Pearson's R = -0.90; p≤0.0.1) (Fig. 4). The slope of this binned C3 plant correlation (i.e., -0.18 ‰ per 100 mm MAP) could be used to estimate the offset of measured plant δ13C values to those expected as a function of MAP. For the binned C3 plants, MAP explained ~82% of the variation in δ13C values."*

Page 16, Line 384-386: "This isotopic contrast corresponds with the vegetation distribution in the basin, with mixed C3 and C4 vegetation in the upper basin and more C3 plants in the lower basin" - Is the vegetational input only controlling factor for the isotopic values? What about any signatures of soil bacteria?

***Reply:*** *Our observation of different soil $\delta^{13}C$ values the upper and lower basin pointed toward a control by the overlying C3/C4 vegetation. As discussed in the response to the reviewer's comment on Page 6, Line 150-152, the contribution of the microbial biomass to the soil organic matter was likely small (i.e., 1-5%), especially during the dry season when the soils were sampled. Regardless, we agree that part of the soil organic carbon would be microbially processed, and potential effects of this degradation process were discussed in section 3.2.1.*

*This now reads: "The Godavari (bulk) soils had on average less negative δ13Corg values in the upper than in the lower basin (-21.4±0.5 ‰, n=22 vs -23.5±0.5 ‰, n=25; p≤0.01) (Kirkels et al., 2021a) (Fig. 3b, 5b). This isotopic contrast corresponds with the vegetation distribution in the basin, with mixed C3 and C4 vegetation in the upper basin and more C3 plants in the lower basin (Fig. 1b, 2). The least negative δ13Corg values were found in soils in the Upper Godavari and North Tributaries (-21.3±0.5 ‰, n=20 and -22.0±0.7 ‰, n=12, respectively) covered by thorny shrublands, dry deciduous forest and predominantly C4 crops, followed by the Middle Godavari (-23.5±0.5 ‰, n=4) in a transition zone, and most negative δ13Corg values were found in soils in the East Tributaries and Lower Godavari that were covered by moist/evergreen forests and C3 crops (-24.7±0.6 ‰, n=6 and -25.1±0.6 ‰, n=5, respectively) (Fig. 1b, 5a,b, S3). These findings correspond with the general observation that the majority of the soil organic carbon derives from microbially processed plant residues,*

*while the microbial biomass itself has been estimated to contribute only 1-5% (Kögel-Knabner, 2002; Simpson et al., 2007). We note that it is challenging to determine actual size of the microbial biomass, which is highly dependent on prevailing moisture levels, availability of easy degradable carbon as energy source and has a high spatial heterogeneity (Birge et al., 2015; Wiesmeier et al., 2019). Given our sampling strategy where soils were collected in the dry season, the low moisture levels likely limited the microbial biomass size and activity in the Godavari soils. Hence, the $\delta^{13}C_{org}$ values in Godavari soils can be interpreted as a time-averaged plant signal on decadal scale that reflects the long-term hydrological conditions that underlie this vegetation distribution."*

Page 18, Line 408-410: "However, C4-derived OC has also been shown to be preferentially incorporated into fine fractions where it is better protected against degradation, whereas C3-derived OC is preferentially added to the coarse fraction thus leaving it less protected" - What governs this affinity for the C4 plants towards finer fractions whereas C3 plants towards coarser fractions?

*__Reply:__ Studies by Wynn (2007) and Wynn and Bird (2007) showed that C4-derived OC would contain more labile, easy degradable compounds. The fine soil particles had presumably a higher ability to stabilise these labile, C4-derived compounds onto mineral surfaces than C3-derived OC that contained more difficult degradable compounds (partially derived from woody plant parts) that preferentially end up in the coarser soil fraction. In addition, the upper basin, where C4 contributions are highest, is characterized by soils that contain more clays than those in the lower basin, and thus facilitated the enhanced association of C4 vegetation with mineral surfaces. We included information on the chemical composition and preferential degradation/stabilisation pathways of C4 and C3-derived OC in section 3.2.1.*

*This now reads: "In mixed C3/C4 ecosystems, C4 plant-derived OC has been shown to contain more labile compounds and thus degrade more rapidly than C3 plant-derived OC that contains more difficult to degrade compounds (Wynn, 2007; Wynn and Bird, 2007). However, C4-derived OC has also been shown to be preferentially incorporated into fine fractions as fine particles are presumed to have a higher ability to stabilise the labile, C4-derived compounds onto mineral surfaces where they are better protected against degradation, whereas C3-derived OC is preferentially added to the coarse fraction thus leaving it less protected (Bird and Pousai, 1997; Wynn, 2007; Wynn and Bird, 2007). In the Godavari basin, Usman et al. (2018) reported similar $\Delta^{14}C$ values for soil OC in the upper and lower basin that have different C3 and C4 plant covers, suggesting that the nett effect of preferential degradation (more young OC) and stabilisation (more old OC) is minor. Indeed, extensive degradation of C4 plant-derived OC is unlikely, given that the upper basin with most C4 plants contains clay-rich, fine particles from weathering of the Deccan basalts (Giosan et al., 2017; Usman et al., 2018; Kirkels et al., 2021b), which would contribute to stabilise the C4-derived OC. "*

References

Basu, S., Agrawal, S., Sanyal, P., Mahato, P., Kumar, S. and Sarkar, A.: Carbon isotopic ratios of modern C3–C4 plants from the Gangetic Plain, India and its implications to paleovegetational reconstruction, Palaeogeogr. , Palaeoclimatol. , Palaeoecol., 440, 22-32, 2015.

Birge, H. E., Conant, R. T., Follett, R. F., Haddix, M. L., Morris, S. J., Snapp, S. S., Wallenstein, M. D. and Paul, E. A.: Soil respiration is not limited by reductions in microbial biomass during long-term soil incubations, Soil Biol. Biochem., 81, 304-310, 2015.

Kögel-Knabner, I.: The macromolecular organic composition of plant and microbial residues as inputs to soil organic matter, Soil Biol. Biochem., 34, 139-162, 2002.

Sankhla, N., Ziegler, H., Vyas, O. P., Stichler, W. and Trimborn, P.: Eco-physiological studies on Indian arid zone plants, Oecologia, 21, 123-129, 1975.

Simpson, A. J., Simpson, M. J., Smith, E. and Kelleher, B. P.: Microbially derived inputs to soil organic matter: are current estimates too low?, Environ. Sci. Technol., 41, 8070-8076, 2007.

Wiesmeier, M., Urbanski, L., Hobley, E., Lang, B., von Lützow, M., Marin-Spiotta, E., van Wesemael, B., Rabot, E., Ließ, M. and Garcia-Franco, N.: Soil organic carbon storage as a key function of soils-A review of drivers and indicators at various scales, Geoderma, 333, 149-162, 2019.

Wynn, J. G.: Carbon isotope fractionation during decomposition of organic matter in soils and paleosols: implications for paleoecological interpretations of paleosols, Palaeogeogr. , Palaeoclimatol. , Palaeoecol., 251, 437-448, 2007.

Wynn, J. G. and Bird, M. I.: C4-derived soil organic carbon decomposes faster than its C3 counterpart in mixed C3/C4 soils, Global Change Biol., 13, 2206-2217, 2007.

Ziegler, H., Batanouny, K. H., Sankhla, N., Vyas, O. P. and Stichler, W.: The photosynthetic pathway types of some desert plants from India, Saudi Arabia, Egypt, and Iraq, Oecologia, 48, 93-99, 1981.

**Point-by-point reply for Reviewer#2 (Sarah Feakins)**

The study is interesting, and would likely be suitable for Biogeosciences after moderate revision.

*We thank the Prof. Feakins for her positive overall assessment of the paper.*

Summary points:

I agree with the other reviewer that I was surprised to find this was a bulk OC carbon isotope based study. In fact, I assumed the study was based on plant wax when I accepted the request to review based upon the paper coming from a biomarker lab. Of course using bulk methods doesn't invalidate the study, but it could be more clearly signaled for the reader. Using the word "bulk" at first reference to on line 32 "In this study we investigated the bulk carbon isotopic signature.." may suffice.

***Reply:*** *This is a good suggestion, we have added the word 'bulk in the abstract, and also specified this throughout the revised MS. It was particularly added in the method section (2.3) describing the* $^{13}C$ *measurements.*

The authors appear to have neglected the changing atmospheric d13C over recent decades and how that would affect carbon in modern plants, and potentially older soils and fluvial SPM. Literature comparisons span 1970s to present and it needs accounting for. Please add discussion of (likely) age of materials and the Suess effect, throughout wherever relevant, and account for this numerically.

***Reply:*** *We are grateful for the reviewers to have pointed this out. We have corrected for the Suess effect by updating the literature data to their modern-day equivalent to allow for direct comparison with the Godavari plant data collected in 2015. For the mixing model and to compare vegetation with soils, riverbed and suspended sediments, we estimated a turnover rate of ~30 years based on literature and corrected the modern Godavari plant data for the Suess effect over this period. An extra table was included to provide an overview of the different values used for comparison (see Table 1 in the revised MS). Correction for the Suess effect in the plant end-members to account for fossil fuel burning over the last decades resulted in a relatively minor change (-2 to +1%) in the estimated %C4 plants, with only a slightly larger reduction (maximal -5%) based on Godavari vegetation without drought correction. Hence, the effect of the Suess correction on the outcomes of the mixing model was relatively small, so our conclusions remain still valid. The overestimation of %C4 due to moisture availability was slightly smaller (6% vs 10%), and there was still a substantial underestimation of the %C4 plants when using global plant end-members compared to those based on Godavari vegetation (15 vs 19%).*

There is some duplication of graphs Figs 1c and 2a, Figs 2b and 3a – that would ideally be organized so that data are only presented only once graphically.

***Reply:*** *We have revised the figures are requested and removed Fig. 1c and 2b.*

Detailed line by line comments follow:

Line 45 "Our analysis revealed that the reconstructed C3/C4 vegetation composition was sensitive to the plant δ13C end-members used as mixing model input." Please do not frame this as a new 'analysis revealed' as it is well known that the C3 'endmember' is a flawed concept as it has a very wide spread. Informed choices about more meaningful endmembers may be possible in some instances e.g. if it is known to be wet rainforest or dry C3 desert for

example. However there have been other attempts to work around this mathematically including the Fwoody cover approach with a nonlinear fit [1].

***Reply:*** *We rephrased Line 45 in the abstract to focus on the results of the C3/C4 reconstruction in the Godavari basin and not frame this as a new concept.*

*This now reads: "Application of a linear mixing model showed that the %C4 plants in the different subbasins was ~7–15 % higher using plant end-members based on measurement of the Godavari vegetation and tailored to local moisture availability than using those derived from data compilations of global vegetation. Including a correction for drought enrichment in Godavari C3 plants resulted in maximal 6 % lower estimated C4 plant cover. Our results from the Godavari basin underline the importance of making informed choices about the plant δ13C end-members for vegetation reconstructions, considering characteristics of the regional vegetation and environmental factors such as MAP in monsoonal regions."*

*We also provided additional motivation for our mixing model approach with respect to the Fwoody cover approach, and included details on the mixing model we used regarding the variance and error propagation in Methods section 2.5.*

*This now reads: "The relative abundance of C3 and C4 plants was estimated based on the isotope mixing model by Philips and Gregg (2001) using linear mass-balance equations and accounting for the variation in the C3 and C4 plants (i.e., sources) as well as in the soils or sediment (i.e., mixture). The δ13CS values were concentration-weighted and error propagation was accounted for. This mixing model provided an estimation of the proportion of C3 and C4 plants, including the standard error of variance on these estimates. Alternative mixing approaches including the C3 fraction woody cover, which accounts for vegetation structure and shading effects (e.g., Wynn and Bird, 2008; Cerling et al., 2011; Garcin et al., 2014), may be complicated by the fact that agricultural use (~60% of the basin) and deforestation since the 19th century have resulted in a more open landscape and has drastically reduced the area covered by native, closed-canopy forests, which is now limited to the East Tributary region."*

*We added a discussion in 3.1 on canopy shading effects for the East Tributary, which is the only subbasin of the Godavari that is still covered by dense forest vegetation.*

*This now reads: "The least negative subbasin-averaged δ13C value for C3 plants was found in the Upper Godavari subbasin that received least precipitation (δ13C: -28.0±0.3 ‰, n=30; MAP: 593±18 mm y-1) , compared to the most negative value in the East Tributaries that received significantly more precipitation (δ13C: -30.1±0.5 ‰, n=5; p≤0.05; MAP: 1530±142 mm y-1; p≤0.01) (Fig. 2, S2). The East Tributaries are the only part of the Godavari basin that is covered by native, wet to moist forests where a denser canopy caused ample shading. This likely resulted in lower soil temperatures and higher moisture and humidity levels in the understory, which generally favours C3 vegetation (Cerling et al., 2011), that was indeed exclusively found in this Godavari subbasin (Fig. 1b, 2). Likewise, Garcin et al. (2014) reported very depleted carbon isotopic signatures for dense tropical forests in Cameroon (>80% tree cover) mainly controlled by water availability, although a 'canopy effect' (van der Merwe and Medina, 1991), which involves recycling of 13C-depleted CO2 in the understory of closed-canopy forests and fractionation due to photosynthesis under low light conditions, may have resulted in additional depletion in the C3 leaves."*

Godavari specific endmembers, this would be more generally interesting if we were told right away if this is the wet or dry end of C3 etc, unlikely that there are regional plant species effects, likely it is just the usual canopy etc effects.

*Reply: We have included information in the abstract on the range of MAP to clarify that the Godavari basin is relatively dry compared to tropical rain forests.*

*This now reads in the abstract: "Our analysis was performed in the Godavari River basin, located in the Core Monsoon Zone in peninsular India, a region that integrates the hydroclimatic and vegetation changes caused by variation in monsoonal strength. The basin has distinct wet and dry seasons and is characterised by natural gradients in soil type (from clay-rich to sandy), precipitation (~500 to 1500 mm y-1) and vegetation type (from mixed C3/C4 to primarily C3) from the upper to the lower basin. Godavari C3 plants confirmed a strong control by Mean Annual Precipitation (MAP) on their δ13C values, with an isotopic enrichment of ~2.2 ‰ from ~1500 to 500 mm y-1."*

*In addition, in the Methods (section 2.4) and Results/Discussion 3.1 we included information on Godavari basin and C3 vegetation in tropical rain forest and their relation to MAP.*

*This reads in 2.4: "Nonetheless, field surveys and data compilations of C3 vegetation in drought-stressed regions reported high inter- and intraspecies variation in C3 plant δ13C values in response to MAP (Ma et al., 2012; Liu et al., 2013, 2014; Basu et al., 2021; Luo et al., 2021). The range of ~500 to 1500 mm y-1 MAP in the Godavari basin was markedly lower than in tropical forests where MAP is typically >2000 mm y-1 and where the majority of global C3 biomass occurs (Kohn, 2010)."*

*This reads in 3.1: "The sampled Godavari plants fell within the typical ranges for global C4 (~−10.5 to –14.5 ‰; Cerling et al., 1997) and C3 vegetation (~-20.5 to –37.5 ‰; Kohn, 2010) (downward corrected by ~-0.5‰ for fossil fuel burning). As in the Godavari basin only a very small area was covered by wet evergreen forest with a MAP of ~1500 – 2000 mm y-1, we observed less negative δ13C values for C3 plants compared to tropical rain forests with MAP typically exceeding 2000 mm y-1 (Kohn, 2010) (Fig. 1b, 2)."*

Line 49 " Hence, incorporating region-specific plant δ13C end-members and drought correction of the C3 end-member in mixing models need to be considered to determine C3 and C4 distributions of modern- and paleo-vegetation in monsoonal regions." Rephrase this sentence.

*Reply: We rephrased this sentence, focusing on the importance of making informed choices about plant end-members.*

*This now reads: "Our results from the Godavari basin underline the importance of making informed choices about plant δ13C end-members for vegetation reconstructions, considering characteristics of the local/regional vegetation and environmental factors such as MAP in monsoonal regions".*

Line 56 – all the cited references refer to bulk plant tissue and are references that span 1970-2010. The difference in plant d13C between 1970 and 2020 is ~2 per mil. Please check and see what a recent collation of data has reported after correction for the date of collection, or do the work to update this to a consistent modern value suitable for comparison to your plants. Your soils and river samples may integrate more time however and thus the temporal shift may also be relevant to summarize here in the introduction.

*Reply: We updated the data in cited references for the Suess effect to modern equivalents for the date we collected the plants samples in the Godavari basin (i.e., 2015), at all places where this was relevant to enable direct comparison. For a good overview, we included the Suess-corrected data in an extra table (Table 1). In Methods section 2.5 we explained the rationale behind this Suess-correction and how we preformed the corrections for the*

*Godavari basin data. In the results and discussion section we always specified how the data were corrected (equivalent to modern or soil age) to enable reliable comparison.*

Line 89 – the concept of endmembers is flawed, especially for C3, instead it is important to describe the spread of C3 plants as context for any central estimate. This section of text is also flawed in that it misses the timescale of sampling. Internal to a study the C4 response to dryness has been found to be quite small 1 per mil (Cerling) not absent as concluded in this plant study in the results section, but perhaps the n is too small to be sure?

***Reply:*** *See our previous answers to Reviewer#1 and Line 45. We expanded information in the introduction about the mixing model approach, including the importance of making informed choices about the plant end-members, also considering their variability as input in the mixing model, and considering the age of the plant samples with respect to the age of the soils or sediments used as mixture in the model and additional influences by environmental factors. In Methods section 2.5 we explained the details of the mixing model approach, including correction for the timescale of sampling. We motivated the choice for a validated model that incorporates the variability in the plant end-members as well as in the soils/sediments, and we accounted for error propagation. In results/discussion section 3.1, we expanded on the influence of drought stress on C4 plant isotopic signatures and reported on contradictory results from other studies, where some found a positive and others no effect by MAP. For the Godavari C4 plants we found no significant effect by MAP, so we inferred that there was no basin to apply a drought-correction for these data.*

Line 94 is on C4, then line 95 returns to C3 again, and another switch is found later on – the flow needs organizing.

***Reply:*** *We reorganised this paragraph to revise the flow. We first focused on C4 and then on C3 vegetation and avoided switching back and forth.*

Line 110 – I do not find the concept of a 'global average' C3 plant d13C to be useful.

***Reply:*** *We reorganised the introduction (and abstract) to focus on the impact of regional conditions (MAP, vegetation structure, occurring species etc.) as relevant influences on the $\delta^{13}C$ signature of plants instead of comparing with global averages. We included a clear description of the concepts 'global' and 'regional' in Methods section 2.5, detailing the difference between end-members based on compilations of globally occurring vegetation versus the measurement of regional plant samples. Also, the impact of the Suess correction is explained in detail.*

*This now reads: "C3 and C4 plant end-members to resolve the mixing model can be based on measurement of regionally occurring, modern vegetation in the Godavari basin (referred to as Godavari-based or regional end-members), which are representative of the prevailing habitat conditions. Alternatively, global end-members can be used based on C3 and C4 plants collected worldwide and reported in literature compilations. Commonly quoted global averages are -27 ‰ for C3 plants (Cerling et al., 1997; Koch, 1998; Dawson et al., 2002) and -12 ‰ for C4 plants (Koch, 1998; Dawson et al., 2002)."*

Line 111 – regional average is also not very useful, more useful to think in terms of the vegetation category average e.g. closed forest, open woodland etc.

***Reply:*** *We introduced the term Godavari-based or regional end-members to refer to $\delta^{13}C$ values of C3 and C4 vegetation that evolve from the specific habitat conditions in the Godavari basin, which likely results from a combined effect of the (Indian) plant species, vegetation structure (density, distribution, canopy etc.), moisture, temperature and other environmental conditions. In method section 2.5, this term is now clearly defined (see also*

previous answer on global average). Unfortunately, our sampling size of the field survey in the Godavari basin did not allow us to identify $\delta^{13}C$ values specific for each vegetation category, and as definitions of vegetation zones may also differ between publications, it was not feasible to find plant $\delta^{13}C$ data to compare each vegetation category to. In the Conclusion we focused on the importance of regional setting to determine representative plant end-members.

Line 116-119 – sentence needs revisiting – rephrase. Note that this refers to a study that is also conceptually based on the endmember approach.

**Reply:** *We rephrased these lines, and included that the study referred to here is based on an end-member approach and that region-specific vegetation was investigated as well as the impact of drought stress.*

*This now reads: "Recently, a study of δ13C values in region-specific vegetation along a precipitation gradient on the Gangetic plain prompted a recalculation of the abundance C3 and C4 plants in sedimentary deposits accounting for drought-stress induced enrichment in C3 plants (Basu et al., 2015, 2019b). They showed that earlier investigations likely underestimated the abundance of C4 plants (~20 %). Recalculation using an end-member and mixing model approach revealed that C4 plants existed in this region at an earlier date than anticipated, changing the timing of (Miocene) C4 grassland expansion on the Gangetic plan to ~17 Ma (Basu et al., 2015, 2019b)".*

Consider moving away from the outdated concept of a C3 endmember and moving to something like the non linear Fwoody cover approach that deals with the issues of spread in C3 plants. Or if you insist upon a linear mixing model make sure you propagate the uncertainties caused by the C3 distribution upon those C4% estimates. If you do error propagation, you'll see the issue.

**Reply:** *See our previous answer to Line 45. In Methods section 2.5, we included all details about the mixing model that we have used. We specify that we used a validated model from Philips and Gregg (2001), which included the variance in the C3 and C4 vegetation (i.e., sources) as well as in the soils and sediments (i.e., mixture), with consideration of the uncertainties in isotope mixing. As described, error propagation was also applied for calculation of the weighted $\delta^{13}C$ values in soils and sediments. We also briefly discussed that application of the Fwoody approach may be complicated by agriculture in the Godavari basin, which has created a more open landscape and drastically limited the area with native forest vegetation. Therefore we argue that variation in the Godavari vegetation d13C cannot be attributed to changes in Fwoody cover, but is driven by changes in moisture availability instead.*

Paragraph beginning 121 discusses plant to soil to river degradation fractionations well. It neglects to discuss the age of the OC and the Suess effect means that 2 per mil needs to be accounted for when comparing today's plants and a couple decade old OC in soil/sediment. Old OC would be 2 per mil more enriched compared to today's OC without any degradation fractionation.

**Reply:** *In the introduction, we included a description that soils and sediments contain a time-integrated $\delta^{13}C$ signal, and consideration of the Suess effect is needed when comparing such values with those in modern vegetation samples. In methods section 2.5 we described how the Suess correction was done, providing an overview in Table 1. Unfortunately, a selection of soils and riverbed sediments analysed by Usman et al. (2018) for the Godavari basin showed too much variation to allow a reliable estimation of the Godavari-specific soil/sediment age, although it did show that there are no significant differences between the*

*ages of soils and sediments, nor between the upper and lower basin. Instead, we used prior studies on savanna ecosystems and tropical forest to estimate a turnover rate of ~30 years.*

*This now reads: "The plant δ13C signal is subsequently transferred to soils or sedimentary deposits, where the δ13Corg signal is assumed to integrate long-term and/or spatial areas and thus incorporate/average the plant δ13C signal for a range of precipitation within this period/region. In order to compare the δ13Corg values of pre-aged soils and sediments with those of modern vegetation a correction for the Suess effect is warranted. Analysis of Δ14C of OC in a selection of Godavari soils and sediments by Usman et al. (2018) revealed no distinct differences between the upper and lower basin nor between bulk soils and riverbed sediments. However, the large variation in Δ14COC values, potentially related to small contributions of very old OC from wind-blown coal dust from the open-pit mines in the north of the basin, made it difficult to determine the average age of OC in Godavari basin. Based on OC turnover rates of ~10 years in tropical forest soils to ~25 – 40 years in savanna ecosystems (Martin et al., 1990; Bird et al., 1996), we estimated an average age of ~30 years for OC in Godavari soils and riverbed sediments. This estimate is at the upper end of recently determined biome-specific OC turnover rates for tropical forests and savannas, where precipitation was shown to have a major effect on soil OC turnover rates (e.g., Carvalhais et al., 2014; Hein et al., 2020). To enable direct comparison of δ13C in plants with δ13Corg in soils and sediments and employ these in the mixing model, we corrected the measured δ13C in modern vegetation for the Suess effect to the average age of soil/sediment OC (i.e., 30 years preceding the plant collection in 2015: 1985) (Table 1)."*

*In the results/discussion (3.2.1), we included a comparison between Suess-corrected plant δ13C values and those in soils when discussing degradation-induced fractionation.*

Line 153 why (paleo-)vegetation reconstructions? "vegetation reconstructions" suffices. Same issue throughout e.g. line 586 and conclusion title line 589.

**Reply:** *Throughout the MS, we changed "(paleo-)vegetation reconstruction" to "vegetation reconstruction".*

Methods

Plant and river sampling methods are appropriate and well described. The only question I'm left with is are the plant samples representative, when sampling bulk from a tree, the trunk is the bulk of the biomass, although the production of leaves may have a faster rate. When sampling leaf wax the leaves are appropriate, but when sampling bulk is the leaf sampling appropriate? I can see it is hard to homogenize a tree unlike sampling grasses (or leaf waxes) where the sampling task is simpler.

**Reply:** *We specified in the revised MS that we sampled the leaves of trees and shrubs and for herbs and grasses we combined leaves and stems. Given that the deciduous trees/shrubs in the Godavari typically shed their leaves annually in the dry season, we argue that leaves are the main contributor to soil OC rather than woody biomass. As for the representation of plant species, we described that we collected the ~5 most dominant species at each site.*

*This now reads: "Samples of above-ground plant material were collected in February/March 2015 (dry season) across the Godavari basin, selecting the 3–5 most dominant species at each site and spanning the full range of plant lifeforms (i.e., trees, shrubs, herbaceous plants and grasses). For shrubs and trees the leaves were collected and for herbs and grasses the leaves and stems were combined. Given that the deciduous trees and shrubs shed their leaves annually in the dry season (Kushwaha and Singh, 2005; Elliott et al., 2006), leaves were considered the main contributor to soil OC rather than woody biomass."*

Line 236 "robust relationship between MAP and d13C has been shown to prevail in C3 plants around the world" yes there is a trend but also a lot of scatter. This is acknowledged on line 249 a long way after for the reader, and the solution we are told is "binning'" on line 249 but binning is not explained, that I have found in the text.

*Reply: We reorganised this section (2.4), as requested. First, we introduced that prior studies found evidence for a relation between MAP and d13C values in C3 plants around the world. We then added information on the amount of scatter in this correlation, including that other studies found high variation in d13C values of C3 plants in response to MAP in drought-stress areas.*

*This now reads: "The Mean Annual Precipitation (MAP) in the Godavari basin was used to evaluate the control of drought stress on plant δ13C values, as prior studies found evidence for a relationship between MAP and δ13C values of C3 plants around the world (Stewart et al., 1995; Diefendorf et al., 2010; Kohn, 2010). Nonetheless, field surveys and data compilations of C3 vegetation in drought-stressed regions reported high inter- and intraspecies variation in C3 plant δ13C values in response to MAP (Ma et al., 2012; Liu et al., 2013, 2014; Basu et al., 2021; Luo et al., 2021)."*

*The binning procedure was applied in order to deal with the uneven distribution of the sampled Godavari C3 plants over the MAP range in the Godavari basin. We added a detailed explanation of the binning procedure in the revised MS in Methods section 2.4.*

*We included the following: "In order to deal with the uneven distribution of Godavari C3 plant $δ^{13}C$ values over the MAP range, we applied a binning approach. The data were binned by calculating the average and standard error of C3 plant $δ^{13}C$ values per MAP range of 100 mm $y^{-1}$. These binned Godavari C3 plant data were subsequently plotted against the average MAP of each bin and utilised for regression analysis to assess the relation between C3 plant $δ^{13}C$ values and MAP."*

Results and Discussion

Line 289 the plants falling in the "lower" end of the global range is consistent with the comparison of modern plants and an older global literature reference comparison. However just on numerical comparison "lower end" also seems to be a misrepresentation as closed tropical forest would be lower. Reconsider.

*Reply: We reconsidered this part and corrected the reported values for the Suess effect, and reported that the Godavari plants fell within the typical ranges for C3 and C4 plants. We included additional information that Godavari C3 plant $δ^{13}C$ values were less negative than those found in closed tropical forests with MAP levels > 2000 mm/y, while wet evergreen forest covers only a very small area in the Godavari basin (East Tributary) that has a typical MAP of 1500-2000 mm/y.*

*This now reads: "The Godavari plants (n=96) showed two distinct groups, with bulk δ13C values that ranged from -12.7 to -15.1 ‰ for C4 plants (n=16, 9 different species) and from -24.3 to -33.2 ‰ for C3 plants (n=77, 38 different species) (Kirkels et al., 2021a) (Fig. 2 3a). The sampled Godavari plants fell within the typical ranges for global C4 (~−10.5 to −14.5 ‰; Cerling et al., 1997) and C3 vegetation (~-20.5 to −37.5 ‰; Kohn, 2010) (downward corrected by ~-0.5‰ for fossil fuel burning). As in the Godavari basin only a very small area was covered by wet evergreen forest with a MAP of ~1500 – 2000 mm y-1, we observed less negative δ13C values for C3 plants compared to tropical rain forests with MAP typically exceeding 2000 mm y-1 (Kohn, 2010) (Fig. 1b, 2)."*

Line 354 – binning – apologies, if I've missed it but I don't see this explained yet, and so I struggle to follow this.

**Reply:** *We included an explanation of the binning approach in Methods section 2.4 (see also our earlier answer for Line 236).*

Line 439 remove "interestingly" which is subjective, and this well-known issue is one reason why reviewer 1 questioned the use of bulk, it becomes problematic in estuarine and marine settings as is well known (and perhaps no longer that interesting).

**Reply:** *We rephrased these lines and removed the word 'interesting', as requested. Furthermore, we added a discussion on how the mixing of riverine and marine OC and their $\delta^{13}C$ composition at the freshwater/seawater interface complicates the tracing of Godavari-derived isotopic signatures to marine sedimentary deposits.*

*This was revised as: ""Notably, δ13Corg values of dry season SPM became less negative near the Godavari's outflows into the Bay of Bengal (Fig. 5c, S3), suggesting mixing of freshwater and estuarine/marine phytoplankton in the delta, where the latter has typically less negative δ13Corg values (i.e., -22.8 to -24.4 ‰). This observation was consistent with changes in electrical conductivity and water isotopic signature (δ18O) that showed seawater intrusion in the delta in the dry season. Regardless, mixing of riverine and marine OC with different carbon isotopic signatures at the outflow complicates the tracing of the Godavari-derived OC signal from the river mouth to marine sedimentary deposits."*

Line 529 – though we found some wood far offshore in the Bengal Fan [2]

**Reply:** *We included a discussion on the fact that no wood particles were found in Holocene, marine sediments in front of the Godavari River mouth, in contrast to the Bay of Bengal Fan where such particles were discovered in sediments spanning the last 19 My. This is likely due to the steep altitudinal gradient covered by the Himalayan-derived Ganges-Brahmaputra River that can carry coarse sediments far offshore at high flow conditions and feeds the Bengal Fan, whereas such a gradient is lacking in the Godavari basin.*

*The following was included in the revised MS: "Indeed, Holocene marine sediments collected in front of the Godavari's mouth contained no woody particles (Ponton et al., 2012; Giosan et al., 2017; Usman et al., 2018), in contrast to the Bay of Bengal Fan fed by the Himalayan-derived Ganges-Brahmaputra, that covered a steep altitudinal gradient and carried coarse sediments far offshore at high flow conditions, where wood particles were found in sediments spanning the last 19 Ma (Lee et al., 2019). "*

Conclusions

Line592 – the discussion makes it sound like there is something regionally unique about the d13C when they fall within the global plants dataset and likely overlap with similar vegetation types. Thus it is more vegetation type/habitat/MAP considerations rather than geographic regions that should be emphasized, and so doing would make it more globally of interest than local.

**Reply:** *We have reconsidered this after application of the Suess-correction. While there are still significant differences between plant end-members determined based on data compilations of global vegetation versus those measured in the Godavari basin, they fall broadly within the same range. Hence, we reframed this concept by focusing on regional settings such as vegetation species, structure, habitat and MAP as factors determining the plant end-members. We have rephrased this throughout the revised MS and we have rewritten the conclusions accordingly.*

Figures

Fig 1 – the map figures are useful, for the third panel showing MAP is the partition of the upper and lower basin based on the MAP, if so or otherwise, please give the numerical basis for the partition in the caption for this panel. Preferably change to a green-brown or blue saturations color scale rather than rainbow to be intuitive visually, and provide a legend that can be read in a quantitative sense, see comment on Fig 2a). Please note the repetition of data visuals, Fig 1c and Fig 2a are duplicative. Duplication should be removed. Fig 1c can be removed, as 2a conveys data at the site sampling points as well as the basemap.

*Reply: We have specified the natural gradients in bedrock geology, precipitation and vegetation type from the upper to the lower basin in Methods section 2.1 in the revised MS. We have removed Fig. 1c, as requested. We changed the colour scheme for MAP to a blue colour saturation scale, as recommended.*

Fig 2 a) apart from other concerns regarding the rainbow color scheme that have been widely reported, I would also not encourage the use of scale bar that is purely qualitative for the MAP data. It is not possible to read between the numbers 430 and 2300 mm/yr and know what 'yellow' or 'green' represents in terms of MAP. You can use a scale with incremental output and a color scheme that is a saturation of a single color which will help to allow for visual quantitative evaluation of where is wetter and drier.

*Reply: The dataset (APHRODITE model by Yatagai et al. (2009)) contains continuous data for MAP and changing to a discrete/incremental output was not feasible within this format. However, changing the colour scheme for MAP to a blue colour saturation scale, as recommended, improved the readability of the scale and helped to better visualise drier and wetter areas within the basin.*

d13C data points with the rainbow colors can be discerned by most readers using the legend, the coloring is not intuitive, for wet to dry try green to brown for example, and it would be better to pick a color scheme that can be seen by all readers.

*Reply: As recommended, we changed the colour scheme for the d13C data points from rainbow colours to saturation colours from white to black, which can be seen by all readers, also by colour blind readers. This saturation colour scheme provides an intuitive colouring from more negative (white) to less negative (black) d13C values.*

b) Why are upper and lower basins parsed. Are these much different, probably not as the C4 distribution in lower basin falls within that for the upper basin, and the same for C3 with the upper basin just having a bit more range. Maybe overlay the two bar charts or use violins, to display the data if you want to keep with this 2 category, but if you do an T or F test do you find they are significantly different? (this panel is repeated in fig 3) fig. 2b can therefore be deleted.

*Reply: As responded on the comment to Fig. 1, the upper and lower basin of the Godavari are distinctly different in terms of underlying geology, C3/C4 vegetation distribution, MAP, and land use types. We have included this information in section 2.1. The upper and lower basin were compared throughout the MS. ANOVA or non-parametric tests were used to compare the upper and lower basin in Fig 3 for significant differences in $\delta^{13}C$ values, and these results were discussed thereafter. The boxplots provided a clear overview of our data, adding or changing to violins would not improve the readability as the spread in data is clearly summarised by the boxes and whiskers, as defined in the figure caption. As requested, we removed Fig. 2b as Fig 3 visualised the information in a more comprehensive way.*

Fig 3 – shows a bar chart of the same data as in figure 2b but in box and whisker format. The data only need to be shown once. As this plot is better this is the plot that should be retained and 2b deleted.

*Reply: We have removed the histogram in Fig 2b and only kept Fig 3, as requested.*

Fig 4 – why show 'global C4' as a line = -12 per mil. Where does this derive from? Is it the mean of a collection of plants over several decades, without representation of the scatter in that dataset or correction for the accelerating d13C change in atmospheric CO2 over the last 2 decades. I assume your plants are simply showing scatter consistent with the global dataset, after correction for atmospheric d13C and pCO2 change over time.

*Reply: We have updated the $\delta^{13}C$ value of C4 vegetation, which derives from compilations of globally occurring vegetation, for the Suess effect to equivalent modern plants (see also Table 1). The cited references refer to the publications were this $\delta^{13}C$ value was mentioned for C4 vegetation. Unfortunately scatter in these data is not reported, which is now discussed in Methods section 2.5. In the Results/discussion we included a note that this value for global C4 plants global and those derived from measurement of Godavari C4 plants generally fall within the same range. We also rephrased this accordingly in the discussion of the mixing model outcomes and the conclusions.*

References

Basu, S., Agrawal, S., Sanyal, P., Mahato, P., Kumar, S. and Sarkar, A.: Carbon isotopic ratios of modern C3–C4 plants from the Gangetic Plain, India and its implications to paleovegetational reconstruction, Palaeogeogr. , Palaeoclimatol. , Palaeoecol., 440, 22-32, 2015.

Basu, S., Ghosh, S. and Sanyal, P.: Spatial heterogeneity in the relationship between precipitation and carbon isotopic discrimination in C3 plants: inferences from a global compilation, Global Planet. Change, 176, 123-131, 2019b.

Giosan, L., Ponton, C., Usman, M., Glusztajn, J., Fuller, D. Q., Galy, V., Haghipour, N., Johnson, J. E., McIntyre, C. and Wacker, L.: Massive erosion in monsoonal central India linked to late Holocene land cover degradation, Earth Surf. Dynam., 5, 781–789, 2017.

Lee, H., Galy, V., Feng, X., Ponton, C., Galy, A., France-Lanord, C. and Feakins, S. J.: Sustained wood burial in the Bengal Fan over the last 19 My, PNAS, 116, 22518-22525, 2019.

Phillips, D. L. and Gregg, J. W.: Uncertainty in source partitioning using stable isotopes, Oecologia, 127, 171-179, 2001.

Ponton, C., Giosan, L., Eglinton, T. I., Fuller, D. Q., Johnson, J. E., Kumar, P. and Collett, T. S.: Holocene aridification of India, Geophys. Res. Lett., 39, L03704-L03709, 2012.

Usman, M. O., Kirkels, F. M. S. A, Zwart, H. M., Basu, S., Ponton, C., Blattmann, T. M., Ploetze, M., Haghipour, N., McIntyre, C. and Peterse, F.: Reconciling drainage and receiving basin signatures of the Godavari River system, Biogeosciences, 15, 3357-3375, 2018.

Wynn, J. G. and Bird, M. I.: Environmental controls on the stable carbon isotopic composition of soil organic carbon: implications for modelling the distribution of C3 and C4 plants, Australia, Tellus B Chem. Phys. Meteorol., 60, 604-621, 2008.

Yatagai, A., Kamiguchi, K., Arakawa, O., Hamada, A., Yasutomi, N. and Kitoh, A.: APHRODITE: Constructing a long-term daily gridded precipitation dataset for Asia based on a dense network of rain gauges, Bull. Am. Meteorol. Soc., 93, 1401-1415, 2012.

---

## Author Response (AR2)

Reply to comments from Editor:

Dear Frédérique and co-authors,

Your revised version addresses the key comments and suggestions raised by both reviewers, and I can therefore accept your manuscript for Biogeosciences pending a few minor corrections and clarifications which I have outlined below.

Best regards
Steven Bouillon

→ *We thank the editor for his positive evaluation of our manuscript and acceptance for publication in Biogeosciences. We address the minor corrections and clarification below.*

-Terminology of how stable isotope ratios are referred to, or how they are compared to each other, needs to be corrected here and there. Some examples below- please give the ms an extra readthrough to check specifically for this.

→ *We checked and corrected this in the revised MS.*

+ L62: "drought enrichment in Godavari C3 plants" : should read something like "an enrichment in 13C of Godavari C3 plants due to drought effects", or "a 13C-enrichment due to drought in Godavari C3 plants"

→ *Corrected, throughout the MS.*

+ L70: "distinct d13C composition" : should be "distinct d13C values", or "distinct stable carbon isotope composition", …

→ *Corrected, throughout the MS.*

+ L71: avoid the use of "signatures"

→ *We replaced "signatures" with (stable carbon isotopic) "values" throughout the MS.*

+ L135: Carbon fractionation : carbon isotope fractionation

→ *Corrected, throughout the MS.*

+ L 176-177: "soil degradation processes enrich OC isotopes" : soil degradation processes enrich the remaining OC in 13C, or : soil degradation processes lead to a 13C-enrichment in the remaining OC, ..

→ *Corrected.*

+ L178: this enrichment : this 13C-enrichment

→ *Corrected, 13C-enrichment and 13C-depletion was specified throughout the MS.*

+ L713: 'more negative, C3-derived OC': the OC is not more negative, its d13C values are more negative, rephrase

→ *Rephrased as: "C3-derived OC with typically more negative δ13Corg values"*

+ L715: depleted : 13C-depleted

→ *Corrected, throughout the MS.*

-section 2.5: you now mention (line 358) that the isotope mixing model is concentration-weighted. Can you clarify if this is correct – if so, you need to mention which C concentrations were used for your end-members.

→ *Concentration-weighted δ13C values were calculated for the soils and sediments in order to deal with variability in OC concentrations of these samples collected in the different subbasins, which varied considerably from ~0.03 to 3.13%. This is now detailed as: "The δ13CS values were concentration-weighted using the TOC content (%) of the individual samples in the (sub)basin". Notably, endmembers are based on plant δ13C values and not affected by this.*

- Regarding the C/N ratios, your Methodology does not specify whether you express these as mass or as molar ratios, please add this information to avoid confusion.

→ *The TOC and TN content were reported as weight% for the soil and sediment samples. This information was added in Methods section 2.3. C/N ratios were reported as mass ratios, this is now also specified in section 2.3 and in the caption of Fig. 6.*

- Also, in Figure 6 you refer to data from Balakrishna and Probst for phytoplankton (C/N ratios between 1 and 8) which seem somewhat implausible – it is highly unlikely that phytoplankton can attain C/N ratios as low as 1 – hence use these values more critically.

→ *Balakrishna and Probs reported that the majority (2/3) of C/N ratios for the Godavari main stem and tributaries fell in the range of 1-8. These samples were mostly collected in the dry season from stagnant to slow moving clear waters composed of fine algal material. However, we agree that only one sample had a C/N ratio <2, and the next lowest C/N ratios equalled 2.5 and 2.6. So we revised Fig 6 and changed the lower limit for phytoplankton C/N ratios to 2.5.*

-L839: C3-domianted : C3-dominated.

→ *Corrected*

-The intro might benefit with some more references on the specific conditions favoring C3 versus C4 plants and their global distribution, for example:
Still et al. (2003) Global distribution of C3 and C4 vegetation: Carbon cycle implications. https://doi.org/10.1029/2001GB001807

→ *Thank you for the suggestion. We added a reference to Still et al. (2003) and references therein in the introduction.*

-Color Figures: thank you for addressing some issues with the color maps. There is some room for improvement though in Figures 3 and 6, these are not accessible to readers with color vision deficiencies as not all symbol-color combinations can be distinguished. This can easily be fixed by using different combinations and/or full versus open symbols.

→ *We thank you for pointing this out. We corrected the colours of Fig 3 and 6 to a palette (Color Universal Design) that is accessible for colourblind readers, following the guidelines described in Katsnelson, A.: Colour me better: fixing figures for colour blindness. Nature 598,*

*224-225, 2021, doi: https://doi.org/10.1038/d41586-021-02696-z. These colours were also applied in Fig 4, to have a consistent colour palette throughout the MS.*